# Phosphate starvation response precedes abscisic acid response under progressive mild drought in plants

Yukari Nagatoshi [1], Kenta Ikazaki [2], Yasufumi Kobayashi [1], Nobuyuki Mizuno [3,10], Ryohei Sugita [4], Yumiko Takebayashi [5], Mikiko Kojima [5], Hitoshi Sakakibara [5,6], Natsuko I. Kobayashi [7], Keitaro Tanoi [7], Kenichiro Fujii [1,11], Junya Baba [1], Eri Ogiso-Tanaka [8,12], Masao Ishimoto [8], Yasuo Yasui [3], Tetsuji Oya [2] & Yasunari Fujita [1,9] ✉

Drought severely damages crop production, even under conditions so mild that the leaves show no signs of wilting. However, it is unclear how field-grown plants respond to mild drought. Here, we show through six years of field trials that ridges are a useful experimental tool to mimic mild drought stress in the field. Mild drought reduces inorganic phosphate levels in the leaves to activate the phosphate starvation response (PSR) in soybean plants in the field. Using *Arabidopsis thaliana* and its mutant plants grown in pots under controlled environments, we demonstrate that PSR occurs before abscisic acid response under progressive mild drought and that PSR plays a crucial role in plant growth under mild drought. Our observations in the field and laboratory using model crop and experimental plants provide insight into the molecular response to mild drought in field-grown plants and the relationship between nutrition and drought stress response.

Since the beginning of the history of agriculture, droughts have threatened crop production around the world. Drought is a complex and unpredictable natural disaster, with varying patterns of intensity, timing, persistence, and frequency[1]. Numerous studies, mainly in *Arabidopsis thaliana* plants, have revealed that the phytohormone abscisic acid (ABA) plays an essential role in regulating the responses to a wide variety of drought stress conditions and in optimizing water use in plants[2,3]. Cellular dehydration increases endogenous ABA levels, which triggers multiple physiological and molecular responses,

including stomatal closure and the activation of dehydration/ABA-responsive gene expression[2,3]. While studies of severe drought stress responses in plants have revealed the underlying molecular mechanisms, such as the ABA signaling pathway, it has recently become clear that different molecular mechanisms are involved in mild or moderate drought stress responses[4–6]. Indeed, not only ABA-related genes but also genes related to photosynthesis, carbohydrate turnover, cell wall expansion, and growth regulation are involved in the response of plants to mild or moderate drought stress conditions[7–12]. However, the

[1]Biological Resources and Post-harvest Division, Japan International Research Center for Agricultural Sciences (JIRCAS), Tsukuba, Ibaraki 305-8686, Japan. [2]Crop, Livestock and Environment Division, JIRCAS, Tsukuba, Ibaraki 305-8686, Japan. [3]Graduate School of Agriculture, Kyoto University, Kyoto, Kyoto 606-8502, Japan. [4]Radioisotope Research Center, Nagoya University, Nagoya, Aichi 464-8602, Japan. [5]RIKEN Center for Sustainable Resource Science, Yokohama, Kanagawa 230-0045, Japan. [6]Graduate School of Bioagricultural Sciences, Nagoya University, Nagoya, Aichi 464-8601, Japan. [7]Graduate School of Agricultural and Life Sciences, The University of Tokyo, Bunkyo, Tokyo 113-8657, Japan. [8]Institute of Crop Science, National Agricultuetre and Food Research Organization (NARO), Tsukuba, Ibaraki 305-8518, Japan. [9]Graduate School of Life Environmental Science, University of Tsukuba, Tsukuba, Ibaraki 305-8572, Japan. [10]Present address: Institute of Crop Science, NARO, Tsukuba, Ibaraki 305-8518, Japan. [11]Present address: Institute of Agrobiological Sciences, NARO, Tsukuba, Ibaraki 305-8604, Japan. [12]Present address: Center for Molecular Biodiversity Research, National Museum of Nature and Science, Tsukuba, Ibaraki 305-0005, Japan. ✉e-mail: yasuf@affrc.go.jp

molecular mechanisms underlying mild drought stress responses of plants in the field, where the environment fluctuates, remain elusive.

Phosphorus is one of the three major nutrients, along with nitrogen and potassium, that condition plant growth in natural soils. Although phosphorus is abundant in many natural soils as organic and inorganic matter, plant roots absorb only inorganic orthophosphate (Pi), which is also essential for microbial growth[13,14] and is rarely present in the soil[14]. Plants have therefore evolved a phosphate starvation response (PSR) system that senses phosphate starvation and regulates root and stem growth accordingly[15]. The genetic basis of the PSR has been intensively studied in the model experimental plant *Arabidopsis*[16]. PSR is controlled by a conserved pathway centered on the SPX receptors and PHR transcription factors[17–19]. PHR transcription factors regulate the expression of PSR genes (e.g., those encoding phosphate transporters, phosphatases, and lipid modification proteins) to enhance Pi uptake under Pi-deficit conditions[20]. The establishment of beneficial relationships with some soil microorganisms is one of the strategies for increasing Pi uptake capacity in plants, and genetic mechanisms of the relationship between the PSR and the immune system are emerging[21,22]. Although it has been reported that drought reduces the amount of Pi in plants[23], the link between the PSR and drought is not well understood.

As a major oil and protein source for food, feed, biodiesel, and industrial products, soybean (*Glycine max* L. Merr.) is the most economically important dicot crop in the world[24]. The growing demand for soybean oil, especially in emerging economies, has led to a rapid increase in soybean production; in recent years, the global production and harvested area of soybean have increased at rates that exceed those of the other major crops[25]. However, since soybean is cultivated mostly under rainfed conditions, soybean production is severely constrained by abiotic stresses, particularly drought[26]. Even mild drought stress that occurs without obvious outward signs of a stress response, such as leaf wilting, poses a major threat to soybean productivity[27]. With this growing importance of soybean and the need for soybean research, techniques have been developed to grow soybean vigorously enough to cross and promote generation in indoor growth chambers with fluorescent lighting, which is commonly used to grow *Arabidopsis* plants[28], and various molecular tools and genomic and mutant resources have accumulated[29]. Thus, soybean is an ideal model crop for bridging the gap between laboratory and field drought stress response mechanism studies, not only because of its economic importance but also because of its well-developed research tools.

A wide range of studies have been conducted in controlled environments, such as indoor growth chambers and greenhouses, and in actual fields, to understand the complex and elusive mechanism of the drought stress response in plants in the field[30]. For example, in indoor growth chambers, mild drought stress tests have been carried out using agar medium containing polyethylene glycol to induce osmotic stress in seedlings, or soil in small pots to provide mild drought over an extended period of time by manually or mechanically controlling and monitoring soil moisture[30,31]. In greenhouses, drought stress tests have been performed to vary irrigation conditions for plants grown in pots, whereas, in the field, the tests have been done in experimental plots with different irrigation conditions and rainout shelters to keep the rain out of the test plots[32]. While it is relatively easy to control the growing environment indoors, it is not easy to mimic the complex drought conditions that occur in the field. Field tests do not have the various constraints that are present in pot tests, but complex changes in the natural environment make it challenging to control environmental conditions such as drought. Although there have been attempts to translate research results from the laboratory to the field, many hurdles remain[33].

Here, using soybean as a model field-grown crop during 6 years of trials, we show that ridges are a useful experimental tool for mimicking mild drought stress in the field. Using this method, we determined that mild drought reduces leaf inorganic phosphate levels and activates the PSR in field-grown soybean plants. Detailed pot-based experiments with field soil indicated that PSR-related gene expression is mainly observed under drought conditions that are too mild to activate ABA-mediated gene expression. Furthermore, using *Arabidopsis* and its mutant plants, in addition to soybean, we demonstrate that the PSR is initiated before the ABA response during mild drought in plants and that the PSR plays a crucial role in plant growth under mild drought. Thus, our findings provide insight into the molecular response to mild drought in field-grown plants and the relationship between nutrition and the plant drought stress response. We also show that a field-to-pot approach, in which a phenomenon observed in the field is examined in pot tests, is an effective strategy for elucidating drought response mechanisms.

## Results

### Ridges induce mild drought and reduce yields in the field

Understanding how plants respond to drought in the field at the molecular level is quite challenging. For example, the commonly used method of setting irrigated and non-irrigated plots makes it difficult to see the effects of drought on the non-irrigated plots when both plots receive sufficient rainfall[34]. Rainout shelters are an effective way to exclude rainfall from the experimental plots, but fixed shelters with permanently closed roofs have unavoidable effects on microclimate, such as changes in temperature and photosynthetic radiation, and while roofs that close only during rainfall can minimize unintended sheltering effects on microclimate, they are very costly to install and maintain[35]. Perhaps in part because of this situation, most drought experiments have been conducted in areas biased toward Europe and the United States[36] and have not been adapted to local conditions and conducted all over the world. While ridges are widely used in crop production for various purposes, including facilitating drainage, they have not been used to explore the drought stress response in plants at the molecular level. Therefore, we focused on "ridges" to overcome the various problems associated with drought trials in the field and to identify molecular mechanisms in drought response in the field. Soybean, which has the highest water use per kilogram of product among staple crops such as maize, wheat, and rice[37], was used as the model for this field experiment.

For each growing season over six consecutive years, we created flat and ridged plots (referred to as "flats" and "ridges", respectively) in a model experimental field to conduct mild drought tests. We monitored the volumetric water content (VWC) of the soil using soil moisture sensors (Fig. 1a, b; Supplementary Figs. 1–3). In 2016, the soil VWC in the flats of our experimental field fluctuated between 34% (−0.17 MPa) and 49% (−0.04 MPa) (Fig. 1c; Supplementary Fig. 4). Although the VWC was strongly affected by rainfall (Fig. 1c), the VWC in ridges (ridge height, 30 cm; Fig. 1a, b) was consistently lower than that of the flats and fluctuated between 27% (−0.53 MPa) and 41% (−0.044 MPa) (Fig. 1c; Supplementary Fig. 4). We compared soybean growth on the flats and ridges. The aboveground biomass of the plants grown on ridges was clearly reduced compared with that of plants grown on flats, even though leaf wilting was not observed in either group of plants (Fig. 1d; Supplementary Fig. 5). Thermal imaging analysis of the soybean plants showed that the leaf temperature of the plants grown on ridges was higher than that of plants grown on flats (Fig. 1e), suggesting that transpiration was lower in the plants grown on ridges, which may have reduced the photosynthetic rate and biomass. Consequently, the yield of soybean grown on ridges was also reduced compared with that of soybean grown on flats (Fig. 1f, g; Supplementary Fig. 6). An increase in leaf temperature and decrease in growth are typical responses of plants to water deficit, suggesting that ridges induce mild drought stress in plants in the field.

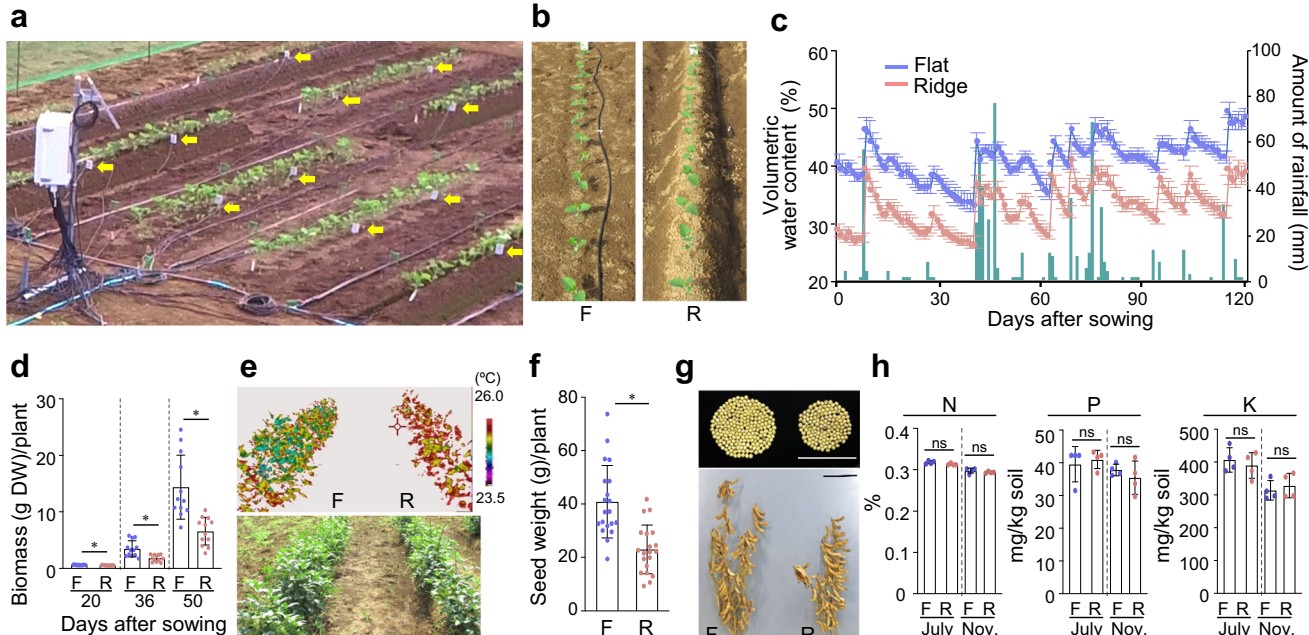

**Fig. 1 | Ridges artificially induce mild drought and reduce soybean yield.**
**a** Photograph of the experimental field, including the VWC data collection station, watering tubes, and probes (yellow arrows). **b** Twelve-day-old soybean plants on flat and ridged plots (referred to as "flats" and "ridges," respectively). F and R denote flats and ridges, respectively. **c** Daily rainfall (green bars) and time course of soil VWC ($n = 4$ independent replicates) in the flats (blue lines and points) and ridges (pink lines and points) over the study period in the 2016 field. Relationships between soil VWC and water potential were 50.7% (−0.0039 MPa, pF 1.6), 44.9% (−0.0098 MPa, pF 2.0), 41.2% (−0.031 MPa, pF 2.5), 38.6% (−0.098 MPa, pF 3.0), and 26.3% (−0.61 MPa, pF 3.8). **d** Aboveground biomass (dry weight, DW) per plant grown on flats and ridges ($n = 12$ biologically independent replicates).
**e** Thermogram and the corresponding digital image of 9-week-old soybean plants grown on flats and ridges in 2016. **f** Total seed weight per plant grown on flats and ridges ($n = 20$ biologically independent replicates). **g** Seeds from individual representative plants grown on flats and ridges. Bars, 10 cm. **h** Nutrient contents (N, P, and K) of soil in flats and ridges before (July) and after (November) soybean cultivation in 2016 ($n = 4$ independent replicates). Error bars in **c**, **d**, **f**, and **h** denote SD. *$P < 0.05$, two-tailed paired samples $t$-test (**d**, **f**). ns, no significant difference, one-way analysis of variance (ANOVA) with Tukey's test in ridged versus flat plots for each season (**h**).

## Irrigation can compensate for the reduced growth caused by ridges

To ascertain whether the reduced growth of the plants on ridges was indeed due to water deficit or rather to changes in the nutrient composition of the soil, we measured the nitrogen (N), phosphorus (P), and potassium (K) contents in the flats and ridges. There was no significant difference in the contents of these nutrients between the flats and ridges, both at the beginning and at the end of the soybean growing season (Fig. 1h). We thus next examined whether watering the ridges would compensate for the negative effect of ridges on plant growth (Fig. 2; Supplementary Fig. 7). Among rainfed plants, there was a noticeable difference in soybean growth on flats and ridges, whereas this difference was reduced or barely detectable among irrigated plants (Fig. 2; Supplementary Fig. 7b–e). Thus, the negative effect of ridges on plant growth was complemented by irrigation, indicating that the reduced growth of the plants on ridges was mainly caused by a water deficit. While a water deficit was not adequately induced on ridges in 2017, the results of the remaining 5 years of field trials demonstrated that the reduction in soybean growth on ridges compared with flats was a result of reduced VWC (Fig. 1; Supplementary Figs. 7–9). Together, these observations demonstrate that ridges are a valuable tool for inducing conditions that mimic mild drought stress in the field.

## PSR is induced in response to mild drought in plants in the field

To examine how mild drought stress affects the expression of genes in soybean plants in the field, we analyzed the fully expanded second trifoliate leaves of 29-day-old pre-flowering plants in the 2015 field experiment using RNA sequencing (RNA-seq) (Fig. 3a, b; Supplementary Fig. 9). We identified 3,045 differentially

expressed genes (DEGs) between the soybean plants grown on flats and those grown on ridges (Supplementary Data 1), which had soil VWCs at sampling time of 41% (−0.017 MPa) and 34% (−0.094 MPa), respectively (Supplementary Figs. 4 and 9). We identified 990 and 400 DEGs that were up- and down-regulated ($|\log_2(FC)| \geq 1$, FPKM value > 0, $q < 0.05$), respectively, under mild drought conditions (Fig. 3c; Supplementary Data 2 and 3). Gene Ontology (GO) enrichment analysis (false discovery rate [FDR] <0.05) revealed that various genes not related to ABA but to nutrition, sugars, vitamins, lignin, and carbohydrates were enriched in the up-regulated DEGs, while no GO terms were significantly enriched among the down-regulated DEGs (Supplementary Data 4). This result is consistent with the fact that there was no clear difference in ABA content between the leaves of soybean plants grown on flats and those of plants grown on ridges (Supplementary Fig. 10). Notably, 16 and 7 of the up-regulated DEGs were related to "cellular response to phosphate starvation" and "phosphate ion homeostasis", respectively (Supplementary Fig. 11). Furthermore, 11.4% of the up-regulated DEGs were PSR genes, including those encoding signaling factors, phosphatases, transporters, and genes implicated in lipid metabolism in *Arabidopsis*[21] and soybean[38,39] (Fig. 3d, e; Supplementary Data 5 and 6), indicating that a battery of PSR genes was up-regulated under mild drought conditions. Mild drought stress induced the expression of PSR marker genes in not only the second but also the first and third trifoliate leaves of soybean plants in the field (Fig. 3f).

## Pi levels decrease in response to mild drought

To determine whether reduced Pi levels indeed occur under mild drought conditions in the field, we measured the Pi concentration of

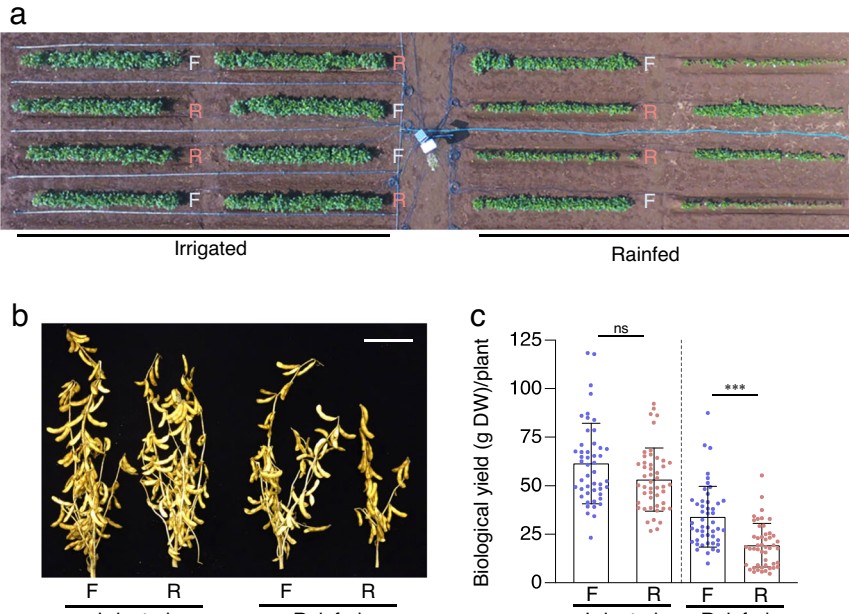

**Fig. 2 | Irrigation compensates for the reduced growth of plants on ridges.**
**a** Aerial view of 7-week-old soybean plants in the 2018 field. F and R denote flats and ridges, respectively. Left half, irrigated area. Right half, rainfed area.
**b** Representative soybean plants from flats and ridges with and without irrigation.

Bar, 10 cm. **c** Biological yield (total dry weight of seed, stem, and pod sheaths) per plant grown on flats and ridges ($n$ = 48 biologically independent replicates). [***]$P$ < 0.001, two-tailed paired samples $t$-test; ns, no significant difference. Error bars denote SD.

the first to third trifoliate leaves. Pi concentrations in leaves at each of these positions of soybean plants grown on ridges were significantly lower than those of plants grown on flats (Fig. 3g). Notably, Pi concentrations in the third trifoliate leaves of soybean plants grown on ridges were higher than those in the first and second trifoliate leaves, and Pi concentrations tended to be inversely related to varying degrees with the expression levels of PSR marker genes (Fig. 3f, g). These results suggest that mild drought stress reduced Pi concentrations in the leaves of field-grown soybean plants, resulting in PSR induction.

We performed elemental analysis using inductively coupled plasma mass spectrometry (ICP-MS) and ICP-optical emission spectrometry (ICP-OES) to determine if other inorganic nutrients were also affected by mild drought (Supplementary Fig. 12). Of the three primary macronutrients, N, P, and K, which are essential for plant growth, only P was markedly lower in the leaves of soybean plants grown on ridges compared with those of soybean plants grown on flats (Supplementary Fig. 12a). Among the secondary macronutrients essential for plant growth, magnesium (Mg) and sulfur (S) contents in the leaves of soybean plants grown on ridges were slightly lower than those of soybean plants grown on flats, but no drastic changes were observed (Supplementary Fig. 12b). Among the micronutrients essential for plant growth, boron (B), zinc (Zn), and nickel (Ni) were significantly higher, whereas copper (Cu) was significantly lower in leaves of soybean plants grown on ridges compared with those of plants grown on flats (Supplementary Fig. 12c). A previous meta-analysis based on 155 observations of plant nitrogen and phosphorus concentrations in a variety of plant species and soils suggested that drought stress has negative effects on plant nitrogen and phosphorus concentrations, with drought conditions inducing a larger decrease of phosphorus than nitrogen[23]. This is consistent with our observation that mild drought conditions reduced P by 47.3% (±7.8%), whereas nitrogen was reduced by only 11.6% (±5.9%) (Supplementary Fig. 12a). These findings support the notion that mild drought stress reduces levels of Pi, among nutrient elements, and induces PSR in plants in the field.

## Mild drought causes PSR, and more severe drought induces ABA response

We next explored if the link between mild drought stress and the PSR identified in our field experiments could be reproduced in pot trials (Fig. 4). When the primary soybean leaves were fully expanded 10 days after sowing in pots containing the same soil as that used in the field experiments, we started drought stress treatments with five different levels of VWC, Water Condition 1 (WC1) to WC5 (increasing water deficit; Fig. 4a). Leaf surface temperatures of 26-day-old soybean plants visualized using infrared thermography paralleled soil VWCs in the pots (Fig. 4a, b), suggesting that the lower the VWC, the greater the reduction in transpiration rate under experimental conditions. We analyzed the gene expression profiles of the fully expanded first trifoliate leaves of 38-day-old seedlings grown in pots. Expression levels of PSR genes were largely inversely associated with VWC levels of WC1 (VWC 42 to 57%, >−0.0039 MPa) to WC4 (VWC 24 to 36%, −0.20 to −0.0035 MPa), but not WC5 (VWC 20 to 29%, −0.50 to −0.056 MPa) treatments (Fig. 4c). Thus, soil water status determines the degree of the PSR under mild drought stress conditions, supporting the findings of our field experiments. Interestingly, even though the VWC decreased between pots WC4 and WC5, PSR gene expression did not increase; by contrast, the expression of ABA-responsive genes, which are established markers of the response to severe drought stress[2,40], increased (Fig. 4c, d). This result was supported by observations in pot tests using vermiculite (Supplementary Data 7), which is commonly used as an artificial soil for experiments, and liquid nutrients with 0.1 mM $KH_2PO_4$. With increasing severity of mild drought (Supplementary Fig. 13a, b), aboveground biomass and leaf water content gradually decreased (Supplementary Fig. 13c, d), whereas the Pi concentration initially decreased but later increased (Supplementary Fig. 13e). By contrast, the ABA content increased as drought exceeded a certain level (Supplementary Fig. 13f). Collectively, these data suggest that PSR occurs mild drought, followed by an ABA response induced by more severe drought (Fig. 4e).

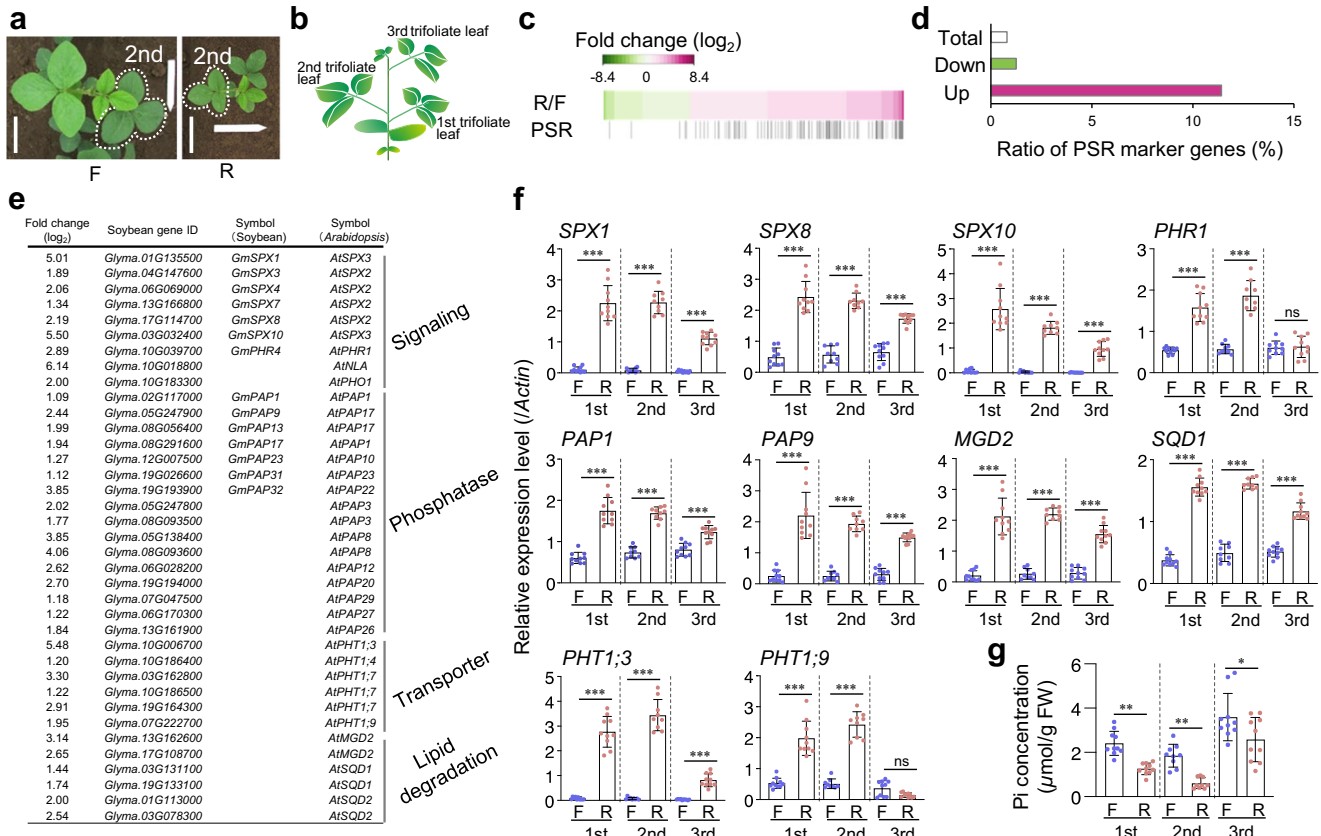

**Fig. 3 | Mild drought induces PSR in field-grown soybean plants. a** Positions of the second trifoliate leaves (dotted enclosure) used for RNA-seq analysis in 29-day-old soybean plants grown on flats and ridges in 2015. Bars, 10 cm. **b** Schematic representation of a soybean plant. **c** Heatmap of $\log_2$ fold-change in expression of 990 up-regulated and 400 down-regulated genes ($|\log_2(FC)| \geq 1$, FPKM value > 0, $q < 0.05$) under mild drought conditions induced by ridges. PSR genes are indicated on the lower row. F and R denote flats and ridges, respectively. **d** Ratio of PSR marker genes to the total number of genes, the number of up-regulated genes, and the number of down-regulated genes. **e** Representative up-regulated PSR marker genes under mild drought conditions. **f** PSR gene expression determined using RT-qPCR in the first to third trifoliate leaves of 29-day-old plants grown on ridges and flats in 2015 ($n = 9$ biologically independent replicates for the second trifoliate leaves; $n = 10$ biologically independent replicates for the first and third trifoliate leaves). The *actin* gene (*Glyma.15G050200*) was used as an internal control for gene expression. **g** Pi concentration of first to third trifoliate leaves of soybean plants grown on ridges and flats; same samples described in **f**. $^{*}P < 0.05$, $^{**}P < 0.01$, and $^{***}P < 0.001$, one-way analysis of variance (ANOVA) with Tukey's test in ridged versus flat plots for each leaf position; ns, no significant difference. Error bars denote SD.

## PSR occurs before ABA response under progressive mild drought

To determine whether the phenomenon of mild drought-causing PSR is independent of plant species, we induce mild drought in *Arabidopsis* plants grown in vermiculite and liquid nutrients with 0.1 mM $KH_2PO_4$. Seven-day-old seedlings grown on agar plates were transplanted into pots filled with vermiculite soaked in water containing liquid nutrients and grown for 6 days. The pots were divided into two groups. Pots containing control plants were soaked in water without nutrients, while pots containing drought-stressed plants were placed on paper towels to reduce soil moisture (Fig. 5a), meaning that the vermiculite in pots of both treatments contained the same amount of nutrients, including phosphate. To further explore the response of plants to water conditions, this experiment also included a rehydration test. After 6 days of drought stress treatment, plants growing in pots that had been soaked in water and left for 1 day were sampled for the rehydration treatment sample. We measured the VWCs of pots and the aboveground biomass of the seedlings after 1, 3, 6, and 7 days of drought stress treatment (Fig. 5b, c). The VWCs of the pots under drought stress conditions decreased gradually and reached approximately 16% (−0.37 MPa) after 7 days of drought treatment; the VWC of pots under control conditions was maintained at around 70% (>−0.0031 MPa) during the experiments. No significant difference in

the aboveground biomass of the seedlings was observed after 1 day of drought treatment, but the biomass was significantly suppressed after 3, 6, and 7 days of drought treatment compared to the control condition (Fig. 5c). Nevertheless, no wilting was seen in any of the plants under the drought stress condition, indicating that this treatment induced mild drought stress in *Arabidopsis* plants grown in pots (Fig. 5d).

We tested whether mild drought stress causes the PSR in *Arabidopsis* plants grown in pots. We performed RNA-seq of the aboveground parts of *Arabidopsis* seedlings in response to mild drought stress and identified 663, 1,158, 1,346, and 871 DEGs ($|\log_2(FC)| \geq 1$, TPM value > 0, $q < 0.05$) after 1, 3, 6, and 7 days of drought treatment (Supplementary Data 8–11), respectively. We observed that 33, 15, 12, and 15% of the up-regulated DEGs after 1, 3, 6, and 7 days of drought treatment were PSR genes, respectively (Fig. 5e; Supplementary Data 6, 8–11), while only 4, 4, 5, and 6% of DEGs were ABA-inducible genes[40], respectively (Fig. 5e; Supplementary Data 8–12). Hierarchical clustering of 2,397 genes that were differentially expressed ($|\log_2(FC)| \geq 1$, TPM value > 0, $q < 0.05$) in response to mild drought stress in at least one sampling in the RNA-seq experiments revealed that two gene sets, designated clusters A and B, of the 19 clusters were enriched in genes involved in biological processes related to the PSR and the ABA response (Fig. 5f; Supplementary Data 13). The genes in

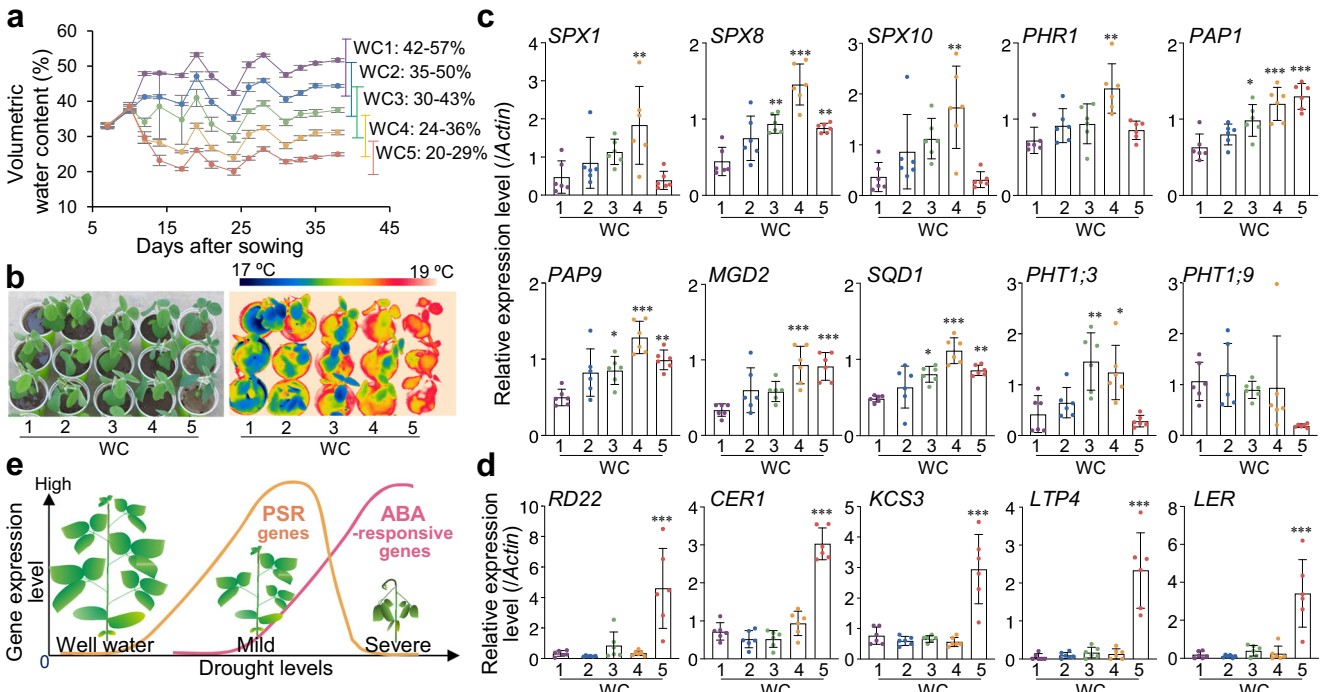

**Fig. 4 | Mild drought causes PSR and severe drought induces ABA response in pot-grown soybean plants. a** Time-course analysis of VWCs in potted soybean plants with five different moisture contents (*n* = 6 independent replicates). The range of variation in VWCs during the period from day 7 after sowing (when plants were transplanted into pots) to day 38 (when sampling was performed) is shown on the right. Drought stress treatments were imposed 10 days after sowing. Relationships between soil VWC and water potential were 36.9% (−0.0039 MPa, pF 1.6), 32.7% (−0.0098 MPa, pF 2.0), 30.0% (−0.031 MPa, pF 2.5), 28.1% (−0.098 MPa, pF 3.0), and 19.1% (−0.61 MPa, pF 3.8). **b** Thermogram and a corresponding digital image of 26-day-old soybean plants in pots. **c, d** Relative expression of PSR (**c**) and ABA-responsive (**d**) genes in the first trifoliate leaves of the 38-day-old plants shown in **b**, as determined by RT-qPCR (*n* = 6 biologically independent replicates). The *actin* gene (*Glyma.15G050200*) was used as an internal control for gene expression. *$P < 0.05$, **$P < 0.01$, and ***$P < 0.001$, one-way analysis of variance (ANOVA) with Tukey's test in WC2–WC5 versus WC1. Error bars denote SD. **e** Proposed model of the plant's response to increasing levels of water deficit stress in terms of gene expression.

cluster A were up-regulated after 1 day of drought treatment, while those in cluster B were up-regulated after 6 days of drought treatment (Fig. 5f, g). Interestingly, cluster A contained 72% of the total PSR genes in the DEGs (97 genes/135 genes), compared to 26% of the total ABA-responsive genes in the DEGs (15 genes/58 genes). In contrast, cluster B contained 64% of the total ABA-responsive genes in the DEGs (37 genes/58 genes), compared to 24% of the total PSR genes in the DEGs (33 genes/135 genes) (Fig. 5f, g). These data show that mild drought stress induces expression of PSR genes before inducing ABA-responsive genes (Fig. 5), consistent with the observation that the Pi concentration gradually decreases during mild drought (Supplementary Fig. 14d), whereas the ABA content increases as drought exceeds a certain level (Supplementary Fig. 14c). Together with the results of mild stress trials in soybean plants grown in pots and the field, these results demonstrate that the PSR induced by Pi reduction is initiated before ABA response in plants grown under progressive mild drought conditions.

**Reduced Pi uptake rather than reduced growth causes PSR**

It remains unclear whether mild drought reduces the Pi concentration in plants by reducing Pi uptake or by inhibiting growth in response to reduced soil moisture. To test whether the reduction in Pi is a consequence of reduced Pi uptake, we performed uptake experiments with radiolabeled $^{32}$Pi (Supplementary Fig. 15a). $^{32}$Pi signals were detected in the aboveground parts of control plants on day 4 but not in the aboveground parts of drought-treated plants (Supplementary Fig. 15b, c), indicating that mild drought inhibits Pi uptake in plants, reducing Pi in plants.

We examined whether the reduction in Pi caused by mild drought was responsible for the reduced growth. Rehydration after 6 days of drought treatment rapidly increased Pi concentration (Supplementary Fig. 14d) and rapidly decreased PSR gene expression (Fig. 5f, g; Supplementary Fig. 14i), but there was no clear difference in aboveground biomass between plants after 7 days of drought treatment and plants rehydrated for 1 day after 6 days of drought treatment (Fig. 5c; Supplementary Fig. 14f). These observations indicate that the reduced Pi due to mild drought is involved in response to soil moisture conditions rather than plant growth. Taken together, these findings support the notion that mild drought inhibits Pi uptake, resulting in reduced Pi and thus inducing PSR in plants.

**PSR plays a crucial role in plant growth under mild drought**

Even when grown with excess Pi (10× Pi: 10 times the amount of Pi), mild drought significantly reduced Pi concentrations compared with those in control plants (Supplementary Fig. 14e), but failed to induce PSR (Supplementary Fig. 14i). Excess Pi promoted plant growth in both control and mild drought-treated plants (Supplementary Fig. 14f–h). These observations suggest that the PSR induced when Pi concentrations in plants fall below a certain threshold affects plant growth under mild drought. Further, to examine the role of PSR under mild drought in plants, we induced mild drought stress in *Arabidopsis phr1 phl1* double mutant plants (Fig. 6; Supplementary Fig. 16), which are deficient in PSR[41,42]. Although there was variation, Pi concentrations were lower under mild drought treatment than under control treatment in both the *phr1 phl1* double mutant and wild-type plants (Fig. 6c, d). Interestingly, when wild-type plants were rewatered, Pi

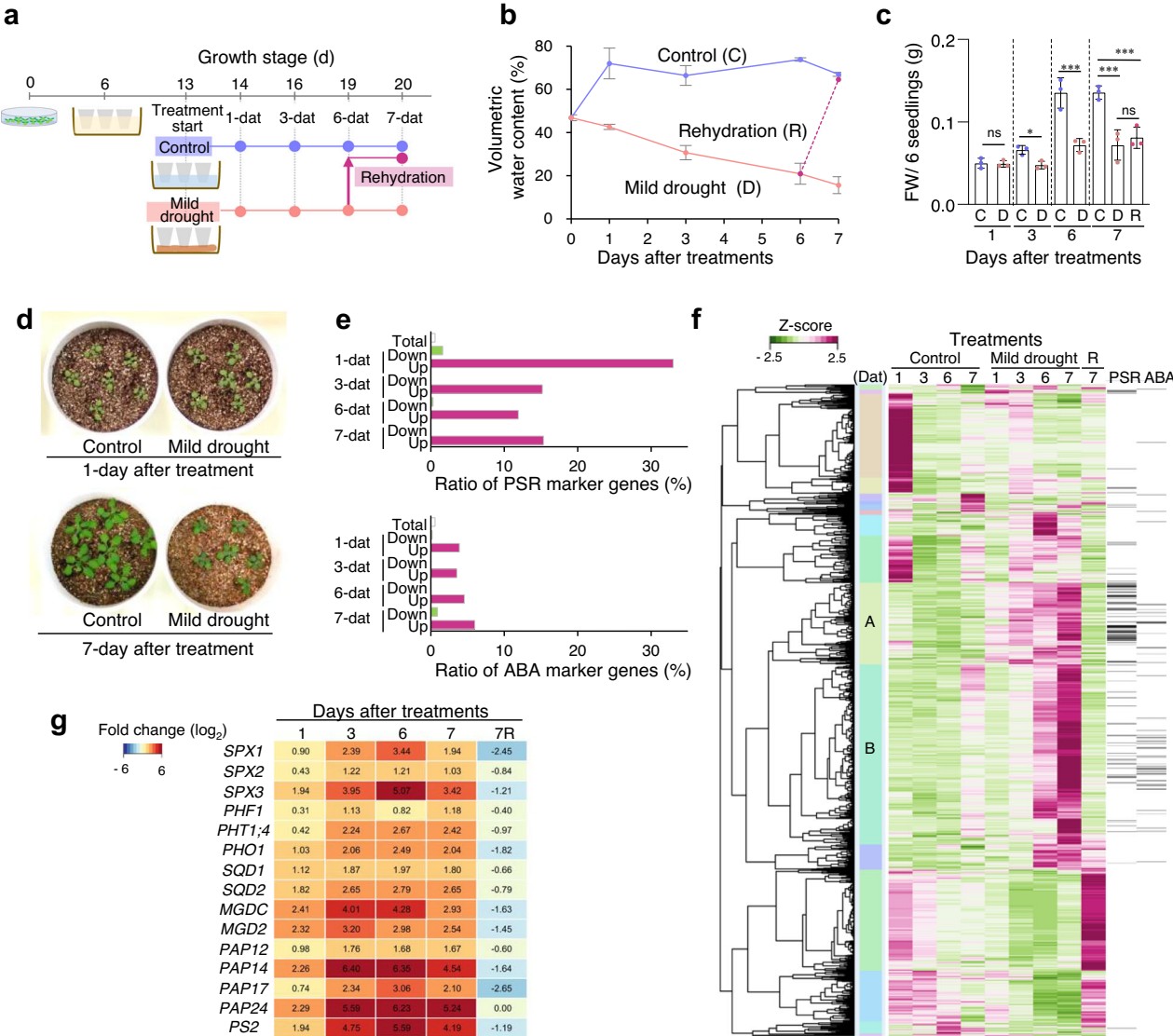

**Fig. 5 | Progressive mild drought induces earlier expression of PSR genes than ABA-responsive genes in *Arabidopsis* plants. a** Schematic overview of the experimental design for control, mild drought, and rehydration in *Arabidopsis* plants. Pots containing 13-day-old plants soaked in water with liquid nutrients after transplanting 6-day-old plants from agar plates were soaked in water as control or placed on paper towels for 6 days for mild drought treatment and then soaked in water for 1 day for rehydration treatment. Plants were sampled at 1, 3, 6, and 7 days after treatment (dat). **b** VWCs at treatment initiation and sampling in pots ($n = 3$ independent replicates). Error bars denote SD. Relationships between the average of soil VWC and water potential were control treatment (69.8%, >−0.0031 MPa), mild drought 1-dat (42.6%, >−0.0031 MPa), mild drought 3-dat (30.7%, −0.019 MPa), mild drought 6-dat (20.9%, −0.21 MPa), and mild drought 7-dat (15.6%, −0.37 MPa). **c** Total aboveground biomass (fresh weight, FW) per pot; six plants were grown per pot under control or mild drought conditions ($n = 3$ independent pot replicates).

*$P <$ 0.05 and ***$P <$ 0.001; ns, not significant; two-tailed Student's *t*-test or one-way ANOVA with Tukey's test (for 7-dat). Error bars denote SD. C, D, and R denote control, mild drought, and rehydration treatment, respectively. **d** Photographs of *Arabidopsis* seedlings at 1 and 7 days after treatment in control and drought-treated pots. **e** Percentage of PSR or ABA marker genes relative to the total number of genes (white bars), the number of up-regulated genes (red bars), and the number of down-regulated genes (green bars) at each sampling time point. **f** Hierarchical clustering of 2,397 genes that were differentially expressed ($|\log_2(FC)| \geq 1$, TPM value > 0, $q < 0.05$) in response to mild drought stress in at least one sampling in the RNA-seq experiments. The right rows indicate PSR and ABA genes. Two gene sets, designated clusters A and B, of the 19 clusters were enriched in genes involved in biological processes related to PSR and ABA responses. **g** Heat maps of $\log_2$ fold-change in expression of representative PSR genes in cluster A.

concentrations increased rapidly, but no significant increase in Pi concentration was observed in the *phr1 phl1* double mutant (Fig. 6c, d), indicating that the PSR is responsible for the rapid increase in Pi concentration during rewatering. To evaluate the effects of the PSR on growth, we measured the aboveground biomass of seedlings and the maximum rosette radius after 1, 3, 6, and 7 days of drought stress treatment (Fig. 6e, f). Both the *phr1 phl1* mutant and wild-type plants grew to varying degrees under control conditions; however, under mild drought conditions, the wild-type plants continued to grow, albeit poorly, while the *phr1 phl1* mutant plants showed no significant

growth (Fig. 6e–h). These data indicate that PSR fulfills a crucial role in plant growth under progressive mild drought.

## Discussion

Here, we show that the PSR occurs prior to the ABA response in plants under progressive mild drought conditions, revealing a novel link between the PSR and drought stress response in plants grown in the field. With the ultimate aim of minimizing crop loss due to insufficiently optimized water supply, this study focused on mild drought stress that is not severe enough to cause the leaves to wilt, in the field,

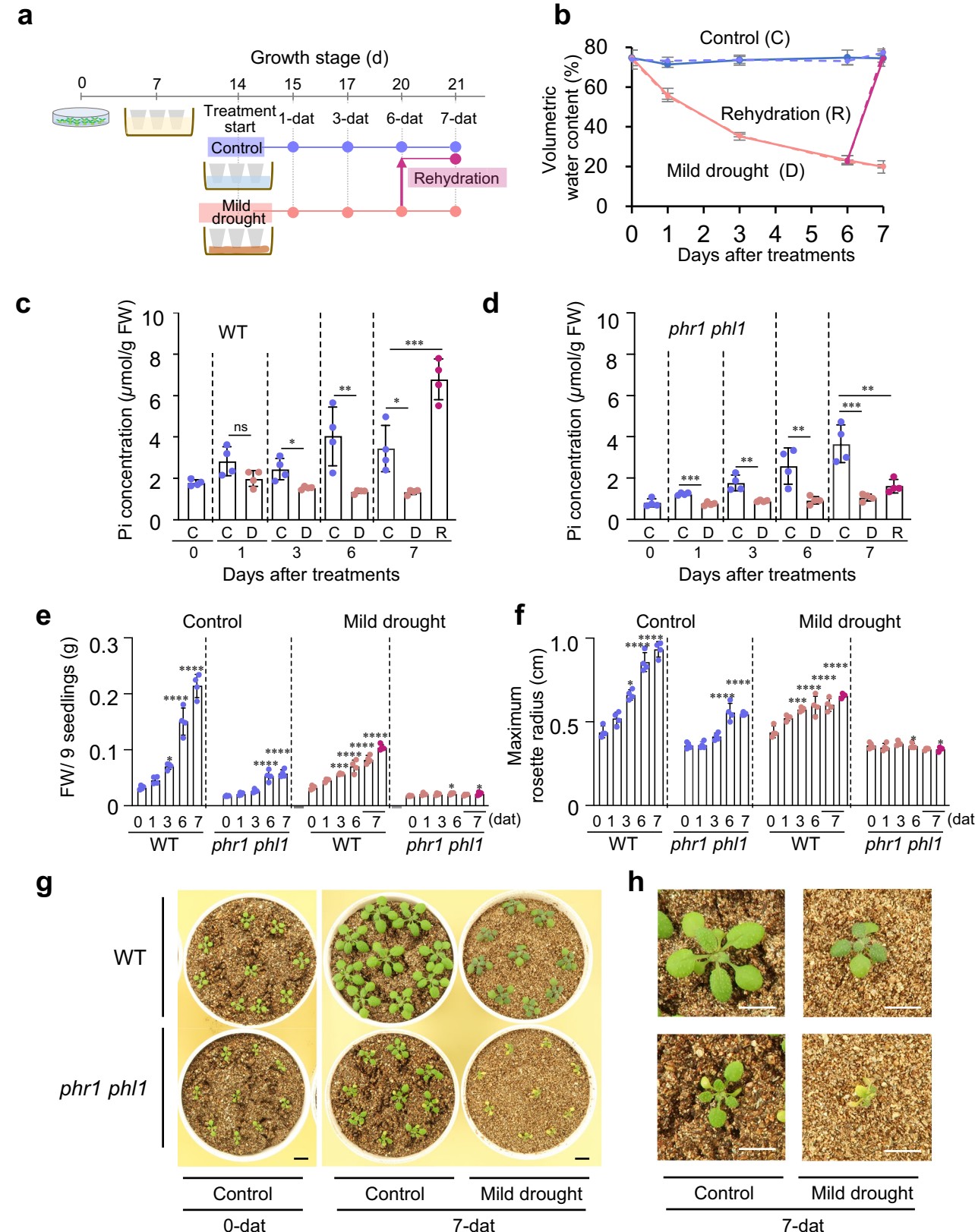

where available P is not abundant and P acquisition is dependent on the biotic and abiotic environment[43]. Previous papers reporting transcriptome analyses of soybean and *Arabidopsis* plants exposed to mild drought stress were unable to detect PSR behavior because, based on leaf wilting and water potential values, these analyses used more severe drought stress conditions and/or more nutrient-rich

conditions in commonly used commercial culture soil than our experiments[7–11,44–46]. We conducted pot tests in the greenhouse using field soil and established that mild drought causes the PSR and that more severe drought causes the ABA response. Moreover, to determine whether such induction of PSR under mild drought is a phenomenon unique to soybean grown in the field, we performed

**Fig. 6 | PSR plays a crucial role in plant growth under mild drought. a** Schematic overview of the experimental design for control, mild drought, and rehydration treatment in *Arabidopsis* plants. Pots containing 14-day-old plants soaked in water with liquid nutrients after transplanting 7-day-old plants from agar plates were soaked in water as control or placed on paper towels for mild drought treatment for 6 days and then soaked in water for 1 day for rehydration treatment. Plants were sampled at 1, 3, 6, and 7 days after treatment (dat). **b** Volumetric water content (VWC) at treatment initiation and sampling in pots ($n = 4$ independent replicates). Error bars denote SD. Relationships between the average of soil VWC and water potential were control treatment (74.0%, >−0.0031 MPa), mild drought 1-dat (55.7%, >−0.0031 MPa), mild drought 3-dat (35.4%, −0.0063 MPa), mild drought 6-dat (22.8%, −0.16 MPa), and mild drought 7-dat (20.0%, −0.24 MPa). **c, d** Pi concentrations of the seedlings in wild-type (**c**) and *phr1 phl1* mutant (**d**) ($n = 4$ biologically

independent pot replicates, three seedlings per pot). Error bars denote SD. $*P < 0.05$, $**P < 0.01$, and $***P < 0.001$, two-tailed Student's *t*-test or one-way ANOVA with Tukey's test (for 7-dat). C, D, and R denote control, mild drought, and rehydration treatment, respectively. **e, f** Total aboveground biomass (**e**) (fresh weight, FW) and maximum rosette radius (**f**); nine plants were grown per pot under control and mild drought conditions ($n = 4$ independent pot replicates). Error bars denote SD. Blue dots, pink dots, and purple dots indicate control, mild drought, and rehydration treatment, respectively. $*P < 0.05$, $**P < 0.01$, $***P < 0.001$, and $****P < 0.0001$, one-way ANOVA with Tukey's test for each day versus 0 dat. **g** Photographs of *Arabidopsis* seedlings at 0 and 7 days after control or mild drought treatment. **h** Enlarged photographs of the central seedlings at 7 days after control or mild drought treatment. Bars, 1 cm (**g, h**).

experiments using *Arabidopsis*, vermiculite, and liquid nutrients and found that PSR gene induction occurs earlier than ABA-induced gene induction under progressive mild drought. Thus, our combined analyses of model crops and experimental plants in the field and in pots in greenhouses and controlled-environment growth rooms, and with the field and artificial soils, provide insights into the mild drought response of plants. Our results, therefore, demonstrate the importance of field-to-lab-oriented research, in which phenomena observed in the field are verified in the lab, to decipher the mechanism underlying the plant's response to mild drought stress in the field. We have also shown that it is crucial to combine laboratory and field studies, using both model crop and experimental plants, to compensate for the shortcomings of either system and use the strengths of both.

We also demonstrate that PSR plays a crucial role in plant growth under mild drought, through the analyses with *Arabidopsis phr1 phl1* mutant deficient in PSR (Fig. 6; Supplementary Fig. 16). Our results with soybean and *Arabidopsis* also show that mild drought inhibits Pi uptake (Supplementary Fig. 15), but that leaf water content is maintained to some extent (Supplementary Fig. 13d), and that Pi concentration in the leaves is reduced (Supplementary Figs. 13e and 14d), thus inducing PSR in plants (Figs. 4, 5; Supplementary Fig. 14i). These results are consistent with the previous observations that reduced soil moisture limits Pi diffusion in the soil[43,47]. Under severe drought conditions, we show that PSR induction is suppressed by a relative increase in Pi concentration (Supplementary Fig. 13e) due to a decrease in leaf water content (Supplementary Fig. 13d). By contrast, ABA cannot respond in situations where some water is maintained in the leaf but responds when the leaf is losing water (Supplementary Fig. 13d, f) and is increasing the osmotic potential[48]; as a result, the PSR occurs prior to the ABA response (Fig. 4e). Although further studies are needed on the molecular mechanisms of PSR in response to mild drought in plants and their relationship to the behavior of intracellular Pi in response to external water status, our findings clearly indicate that PSR genes, whose expression is induced earlier than ABA-responsive genes, are potentially promising early indicators of mild drought stress in plants. So far, the optimal moisture supply to plants has been calculated by changes in soil moisture contents, which are a feature of the external environment surrounding plants, and by estimating transpiration through analysis of plant surface temperature. However, because appropriate soil moisture conditions for plants vary with plant condition and growth and soil type[49], and because it is difficult to eliminate the effects of solar radiation and wind speed in the field using thermographic analysis[50], there has been a need for methods to properly determine plant moisture stress conditions in the field. Since in addition to ABA-responsive gene expression, the PSR gene expression could be used as an indicator of how plants perceive external moisture, the use of such gene markers, combined effectively with soil moisture and thermographic data, may enable more accurate detection of mild drought stress in plants and optimization of irrigation before field crop yields are affected by mild drought stress.

Our consecutive 6-year field trials also demonstrate that ridges are a useful experimental tool for inducing mild drought stress in the field. While ridges are widely used in crop production for various purposes, they have not been used to study drought stress responses in plants. Our research using ridges provides insight into the link between plant nutritional and drought stress responses and paves the way for elucidating the molecular mechanism of the drought response in field-grown plants. Since it is easy and inexpensive to create ridges in experimental fields, this method of inducing mild drought stress, albeit varies in degree depending on rainfall and soil type, overcomes a major research barrier to deciphering the response of field-grown crops to various degrees of drought stress and is expected to facilitate efforts around the world, even in countries with emerging economies where it may not be possible to build expensive facilities. The effects of mild drought are difficult to determine visually in the field, but they significantly reduce yield, so proper diagnosis is critical for crop development and water management to maintain yields. Global food security depends on producing crops that can maintain productivity even under mild to moderate soil water deficits[51,52] and fertilizer deficiencies[53]. Thus, our method of inducing mild drought using ridges and our identification of the PSR as a novel marker of mild drought stress constitute an important basis for improving future food security.

## Methods
### Field experiment
Soybean (*Glycine max* L. Merr.) 'Williams 82', a sequenced model US cultivar[54], was used in this study in all years other than 2019. Soybean 'Enrei', a sequenced model Japanese cultivar[55], was used only in the 2019 field experiment. Field experiments were conducted at the experimental field of the Japan International Research Center for Agricultural Sciences (JIRCAS, 36°03′14″ N, 140°04′46″ E, 24 m above sea level) located in Tsukuba, Ibaraki, Japan, from 2015 to 2020. No fertilizers were applied to the experimental field. To mimic mild drought stress in the experimental field, ridges were created using a rotary tiller after plowing (Supplementary Fig. 1a). In field trials conducted in the 2015–2018 growing seasons, ridges were produced using a TA6 or TA800N tiller (Kubota, Osaka, Japan) coupled to an RA3 rotary (Kubota) and were 30 cm high and 38 cm wide at the base. In the 2019 and 2020 growing seasons, since the same size rotary was unavailable, ridges were created using a TA800N tiller (Kubota) coupled to an SKB-E15C rotary (Kubota) and were 30 cm high and 63 cm wide at the base. The thickness of A horizon, a topsoil characterized by organic matter accumulation, is greater than 30 cm in the experimental field, so both flat and ridged plots were composed of this topsoil.

In 2015, the experimental layout consisted of two areas with flat and ridged plots (Supplementary Fig. 2a). In each area, three plots were used for analysis. Soybean seeds were sown at 20-cm intervals on August 3rd, and 30 plants were grown per plot. For 5 days after sowing, water was sprayed every day to prompt seed germination. In 2016, the experiment was laid out in a randomized complete block design,

including flat and ridged plots, with four blocks for four replicates (Supplementary Fig. 2b). Soybean seeds were germinated on vermiculite (grain size: 3–5 mm)[28] in a temperature-controlled phytotron at JIRCAS (27 ± 6 °C), on July 7th, and 30 healthy and uniform 6-day-old seedlings were planted per plot at 20-cm intervals on July 13th. In 2017, the experimental layout was divided into two areas, one irrigated and the other rainfed. Each area was composed of randomized blocks, including flat and ridged plots, with four replicates (Supplementary Fig. 3a). In the irrigated area, EVAFLOW irrigation tubes (Mitsubishi Chemical Agri Dream, Tokyo, Japan) were installed to supply water to the entire area of each plot. Soybean seeds were sown at 20-cm intervals on July 14th, and 15 plants were grown per plot. To avoid drought stress during seedling establishment, all plots, including non-irrigated plots, were surrounded by irrigation tubes and irrigated four times from the day before sowing to 7 days after sowing. Subsequently, the irrigated area was irrigated 20 times during the period from 18 to 95 days after sowing. Too much irrigation may have increased soil moisture throughout the test area and failed to induce adequate water deficit on ridges (Supplementary Fig. 8a). In 2018, the experimental layout consisted of two areas, one irrigated and the other rainfed (Supplementary Fig. 2c). Each area comprised randomized blocks, with flat and ridged plots, and with four replicates. In the irrigated area, each plot was surrounded by irrigation tubes and irrigated nine times during the period from 15 to 53 days after sowing (Supplementary Fig. 7a). Soybean seeds were sown at 20-cm intervals on July 19th, and 25 plants were grown per plot. In 2019, the experiment was laid out in a randomized complete block design, including flat and ridged plots, with four blocks for four replicates (Supplementary Fig. 3b). Seeds were germinated on vermiculite in a temperature-controlled phytotron (27 ± 6 °C) on July 12th, and 20 healthy and uniform 6-day-old seedlings were planted per plot at 20-cm intervals. In 2020, the experimental layout consisted of two areas, one irrigated and the other rainfed (Supplementary Fig. 3c). Each area comprised randomized blocks, with flat and ridged plots, and with four replicates. In the irrigated area, each plot was surrounded by irrigation tubes and irrigated only once, at 28 days after sowing (Supplementary Fig. 8c). Seeds were germinated on vermiculite in a temperature-controlled phytotron (27 ± 6 °C) on July 30th, and 25 healthy and uniform seedlings were planted per plot at 20-cm intervals on August 6th.

### Measurements of soil water contents in the field

Based on a preliminary investigation of soybean roots in the flat and ridged plots in the experimental field of JIRCAS, soil moisture content was measured at 0–20 cm or 0–30 cm, starting from the top of the ridge or the flat, in this study. In 2015, the volumetric water content (VWC, m³ m⁻³) at a depth of 0–20 cm was measured in each experimental plot at intervals of approximately 3 days using a Hydrosense II 20-cm-long time domain reflectometry (TDR) probe (Campbell Scientific Inc., UT, USA). In situ calibration was conducted using the gravimetric method according to the manufacturer's protocol. Tentative VWC values were measured using the TDR probe, and soils at the measured sites were collected from a depth of 0–20 cm using polyvinyl chloride pipes (4.3 cm in diameter and 20 cm in length). Soil weights including water were determined. After pre-drying at 60 °C for more than 24 h, soil samples were removed from the pipes and oven-dried at 105 °C for more than 24 h to calculate the gravimetric water content (g g⁻¹). A total of eight points were measured. Gravimetric water content was then converted into VWC by multiplying it by the bulk density (BD, Mg m⁻³), assuming that water density is 1.0 Mg m⁻³. Since Andosols generally have a high porosity[56] (about 70% in this study) and can be easily compressed, it was difficult to collect an accurate volume of soil from a depth of 0–20 cm using a single pipe in the field. Therefore, to determine the BD of soils from a depth of 0–20 cm, soil samples were carefully collected from a depth of 10–15 cm using 100-ml soil sampling cores (DIK-1801-11; Daiki Rika

Kogyo, Saitama, Japan) at 12 points, and dry weights of the 100-ml soil samples were then determined after oven-drying at 105 °C for more than 48 h. The average BD of soils from a depth of 0–20 cm was 0.734 ± 0.017 (mean ± SD) Mg m⁻³. From the relationships between the tentative VWC values recorded by the TDR probe (estimated value) and those determined in situ (true value), a calibration formula was obtained: actual VWC = 0.9718 × tentative VWC + 0.1255.

In 2016–2020, the VWC at a depth of 0–30 cm was automatically recorded in each plot every day at 10:00 a.m. using a CS616 30-cm-long TDR probe (Campbell Scientific, Inc.) connected to a CR1000 data logger (Campbell Scientific, Inc.). Soil temperature at a depth of 12–18 cm was also recorded with a model 107 temperature probe (Campbell Scientific, Inc.) to correct for the temperature dependence of the TDR probes, according to the manufacturer's protocol. Estimated VWCs recorded by the TDR probes were similarly calibrated using the gravimetric method according to the manufacturer's protocol. Eight TDR probes were selected for calibration and inserted into plots with temperature probes in the experimental field. Soil samples were collected from four points of the plot using polyvinyl chloride pipes (4.3 cm in diameter and 30 cm in length), and soil weight was determined. Dry weights of the soil samples were determined as described above. BD was measured at a depth of 0–10 cm and 10–30 cm to determine the BD at a depth of 0–30 cm because the BD differed at each depth under test conditions in the field. To determine the BD of soils at a depth of 0–30 cm, soil samples were carefully collected from depths of 3–8 cm and 20–25 cm using 100-ml soil sampling cores (DIK-1801-11; Daiki Rika Kogyo) at eight points each. The BD for depths of 0–30 cm was determined to be 0.773 Mg m⁻³ from the combined weighted average of the values obtained for depths of 0–10 cm and 10–30 cm. The output period data for each TDR probe at the sampling times were corrected for temperature, and tentative VWCs were calculated according to the manufacturer's protocol. Using the averages of tentative VWC values obtained using the eight probes and the corresponding actual VWCs determined in situ at each sampling point, the following calibration formula was generated: actual VWC = 0.9397 × tentative VWC + 0.1638.

### Soil water retention curve

For the field experiments, soil samples were collected from a depth of 10–15 cm using 100-ml soil sampling cores (DIK-1801-11; Daiki Rika Kogyo) at six points in each plot. The soil water retention, a relationship between the gravimetric water content and soil water potential ($\psi$), of each sample was measured using a pressure plate apparatus (DIK-3404; Daiki Rika Kogyo)[57] at a water potential of between −0.0039 and −0.098 MPa, and with a dew point potentiometer (WP4C, Decagon Devices)[58] at values below −0.31 MPa. VWC ($\theta$) values at depths of 0–20 cm and 0–30 cm were calculated using BD values at each depth, assuming a water density of 1.0 Mg m⁻³. Averaged VWCs are shown at matric potential values of −0.0039, −0.0098, −0.031, −0.098, −0.619, and −1.554 MPa (Supplementary Fig. 4). Regression curves at depths of 0–20 cm and 0–30 cm were obtained by fitting the Fredlund and Xing model[59] using SWRC Fit:[60]

$$\theta = \theta_s \left[ \frac{1}{\ln\left[ e + (\psi/a)^n \right]} \right]^m$$

where $\theta_s$ is the saturated VWC, $e$ is Napier's constant, and $a$, $n$, and $m$ are three different soil parameters. The model showed a good fit ($R^2 = 0.985$, $P < 0.001$ for both curves).

For the pot experiments with the field soil (BD value: 0.563), the soil water retention obtained above was used. For those with two types of horticultural vermiculite (small and medium grain), a relationship between the gravimetric water content and soil water potential was similarly measured and VWC ($\theta$) values were calculated using BD values (0.33 for small grain and 0.14 for medium grain), assuming a

water density of $1.0\,Mg\,m^{-3}$. VWCs at each matric potential value are shown (Data Source). Regression curves for medium and small grains were obtained by fitting the Fredlund and Xing model and below the dual-van Genuchten model[61], respectively, using SWRC Fit:[60]

$$\theta = \theta_s \left[ w \left[ \frac{1}{1 + (\alpha_1 \psi)^{n_1}} \right]^{m_1} + (1 - w) \left[ \frac{1}{1 + (\alpha_2 \psi)^{n_2}} \right]^{m_2} \right]$$

$$m_1 = 1 - 1/n_1, \, m_2 = 1 - 1/n_2$$

where $w$ and $\alpha$ are soil parameters. The models showed a good fit ($R^2 = 0.998$, $p < 0.001$ for both curves).

## Measurements of soil nutrients

Soil samples were taken from each plot at a depth of approximately 0–30 cm using a polyvinyl chloride cylinder or soil sampler (DIK-1601; Daiki Rika Kogyo) on July 7 and November 17, 2016, which corresponded to the start and end of the soybean cultivation period, respectively. Samples were air-dried and passed through a 2-mm sieve, and the fine earth obtained was subjected to soil chemical analysis. The oven-dried weight of the fine earth was measured to obtain moisture correction factors that were used to convert nutrient contents of air-dried soil to a dry weight basis.

Total carbon and total nitrogen contents were determined using the dry combustion method with an elemental analyzer (Sumigraph NC-220F; Sumika Chemical Analysis Service, Osaka, Japan). Exchangeable bases ($K^+$) extracted in 1 M ammonium acetate (pH 7) were measured with an inductively coupled plasma atomic emission spectroscopy (ICPE-9000 spectrometer, Shimadzu Corporation, Kyoto, Japan)[62]. Available phosphorus was determined using the Bray-II method[63] with a UV-1800 spectrophotometer (Shimadzu Corporation).

## Meteorological data

Meteorological data, such as the amount of rainfall, were obtained from the Weather Data Acquisition System of the Institute for Agro-Environmental Sciences, National Agriculture and Food Research Organization (http://www.naro.affrc.go.jp/org/niaes/aws/). The weather station is located at 36°01′ N, 140°07′ E, approximately 5 km from the JIRCAS experimental field.

## Evaluation of growth and yield performance of soybean in the field

To investigate the effect of mild drought stress induced by ridges on soybean growth, the aboveground biomass was measured at three-time points per growing season in 2015 and 2016. In 2015, five plants (one or two plants per plot), seven plants (two or three plants per plot), and seven plants (two or three plants per plot) were selected randomly for measuring the aboveground biomass at 21, 30, and 44 days after sowing, respectively. In 2016, 12 plants (three plants per plot) were selected randomly for measurements of aboveground biomass at 20, 36, and 50 days after sowing. The dry weight of the aboveground biomass was measured after oven-drying at 65 °C for more than 48 h.

The yield of soybean plants in field trials was evaluated during 2015–2020. In 2015, 11 plants (three to four plants per plot) grown on the flat plots and 12 plants (four plants per plot) grown on the ridged plots were selected randomly and harvested on December 1st. The number of seeds per plant and total air-dried seed weight per plant were measured. In 2016, 20 plants (five plants per plot) for each condition were selected and harvested on November 10th. After oven-drying at 65 °C for more than 48 h, the following parameters were measured for each plant: number of seeds, total seed weight, stem weight, and pod sheath weight. Harvest index was calculated as total seed weight/(total seed weight + stem weight + pod sheath weight). In

2017, 24 plants (six plants per plot) were selected randomly for each treatment and harvested on November 15th. After oven-drying at 65 °C for more than 48 h, the following parameters were measured for each plant: number of seeds, total seed weight, combined stem and pod sheath weight, and plant height. In 2018, 48 plants (12 plants per plot) were selected randomly for each treatment and harvested on November 14th. After oven-drying at 65 °C for more than 48 h, the following parameters were measured for each plant: number of seeds, total seed weight, stem and pod sheath weight, plant height, and total branch length. In 2019, 40 plants (10 plants per plot) were selected randomly for each treatment and harvested on November 13th. Since soybean plants were damaged by a typhoon on September 8th, only the total biomass, including seeds, stem, and pod sheath, was measured as preliminary data. In 2020, 40 plants (10 plants per plot) were selected randomly for each treatment and harvested on November 2nd. After oven-drying at 65 °C for more than 48 h, the following parameters were measured for each plant: total seed weight and combined stem and pod sheath weight.

## Image acquisition

Images of soybean plants were taken using a CPA-T640A thermal imaging infrared camera (CHINO, Tokyo, Japan) according to the manufacturer's protocol. Aerial images of the experimental field were obtained using a Phantom 4 unmanned aerial vehicle (DJI, Shenzhen, China) according to the manufacturer's protocol.

## Transcriptome analysis of soybean plants grown in the field

Ten plants (three or four plants per plot) were randomly selected from each treatment group, and the first to third trifoliate leaves were collected individually and immediately frozen in liquid nitrogen in the field on September 1, 2015. The frozen second trifoliate leaves were powdered using a multi-bead shocker (Yasui Kikai, Osaka, Japan), and the other leaves were powdered using a mortar and pestle. For RNA-seq analysis, total RNA was extracted from powdered samples of fully expanded second trifoliate leaves using RNeasy Plant Mini Kit (Qiagen) following the manufacturer's protocol. Total RNA samples extracted from three independent plants per plot were mixed for one replicate, and three biological replicates were performed for each treatment. The extracted RNA was used to construct paired-end libraries using a TruSeq RNA Library Preparation Kit v2 (Illumina). The libraries were sequenced on an Illumina HiSeq 4000 sequencer with a 100-bp paired-end protocol (Macrogen). Trimmomatic v0.36[64] with the options SLIDINGWINDOW:4:20, MINLEN:40, LEADING:20, and TRAILING:20 was used for trimming low-quality reads and adaptor sequences. Trimmed reads were mapped against the reference sequence of 'Williams 82' (Wm82.a2.v1: https://phytozome.jgi.doe.gov/pz/portal.html) using STAR v2.5.1[65] with the options outSJfilterReads: Unique; outFilterMatchNminOverLread: 0.96; and outFilterScoreMinOverLread: 0.8. The FPKM (fragments per kilobase of exon per million mapped reads) of each gene was calculated using Cufflinks v2.2.1[66]. Differential expression analysis was performed using Cuffdiff embedded in Cufflinks v2.2.1[66]. Genes that showed a $q$-value < 0.05 were defined as differentially expressed genes (DEGs) between samples grown on flat and ridged plots. Gene annotation such as homology to *Arabidopsis* genes and GO was obtained from gene annotation information in Wm82.a2.v1 (Gmax_275_Wm82.a2.v1.annotation_info.txt).

A heatmap of up- and down-regulated DEGs was constructed using the 'heatmap.2' function of the gplots package v3.1.3 in R software v3.3.3[67]. Representatives of the different types of phosphate starvation–responsive genes were selected among up-regulated genes based on previous reports[21,38,39]. GO enrichment analysis using the clusterProfiler package v2.99.2 in R[68] was performed for the up- and down-regulated DEGs, satisfying the following criteria: $|\log_2(FC)| \geq 1$, FPKM value > 0, $q < 0.05$. A directed acyclic graph (DAG) was constructed using the clusterProfiler

package v2.99.2 in R. GO terms were regarded as significantly enriched when FDR < 0.05. GO descriptions were obtained using AnnotationHub ("AH13355") in R[69].

## ABA quantification

Powdered and frozen soybean samples were distributed into 2-ml tubes, freeze-dried using an FDU-2200 freeze dryer (EYELA, Tokyorika, Tokyo, Japan), and weighed (dry weight). For *Arabidopsis*, three seedlings from each pot were collected in 2-ml tubes, immediately weighed to determine fresh weight, and frozen in liquid nitrogen. ABA was extracted, semi-purified[70,71], and quantified using an ultra-high-performance liquid chromatography (UHPLC) electrospray interface quadrupole orbitrap mass spectrometer (UHPLC/Q-Exactive™; Thermo Scientific) with an ODS column (AQUITY UPLC BEH C18 1.7 μm, 2.1 × 100 mm, Waters)[71,72]. ABA was separated with linear gradients of solvent A (0.06% acetic acid) and solvent B (0.01% acetonitrile) set according to the following profile; 0–10 min, flow rate of 0.25 mL min$^{-1}$, 5–25% B; 10–11 min, flow rate of 0.25 mL min$^{-1}$, 25–29.5% B; 11–15 min, flow rate of 0.25 mL min$^{-1}$, 29.5–35.5% B; 15–15.1 min, flow rate of 0.25 mL min$^{-1}$, 35.5–99% B; 15.2–16.999 min, flow rate of 0.4 mL min$^{-1}$, 1% B; 16.999–17 min, flow rate of 0.25 mL min$^{-1}$, 1–5% B; 17–21 min, flow rate of 0.25 mL min$^{-1}$, 5% B. The column temperature was maintained at 50˚C. Target MS (*m/z*): 263.12888 for unlabeled ABA and 269.16654 for [2H6] ABA. The mass spectrometer was operated under an ESI negative mode with a targeted-selected ion monitoring scan followed by data dependent-MS/MS scans mode (t-SIM/dd-MS2). t-SIM was used for the quantification at resolution 70,000. The automatic gain control target was set at $5 \times 10^4$ ions, and the maximum ion injection time was at 250 msec. Source ionization parameters were optimized with the spray voltage at −2.5 kV, and mass tolerance was set at 5 ppm. Other parameters were as follows: transfer temperature, 200 °C; S-lens level, 50; heater temperature, 400 °C; sheath gas, 40; aux gas, 10. Data were processed by XCalibur TM 4.2.47 (Thermo Fisher Scientific).

## RT-qPCR analysis

Total RNA was extracted from soybean leaves and *Arabidopsis* seedlings using RNeasy Plant Mini Kit (Qiagen) or RNAiso Plus (Takara Bio) according to the manufacturer's protocols. Extracted total RNA was treated with RQ1 RNase-free DNase (Promega). Complementary DNA (cDNA) was synthesized using PrimeScript RT Master Mix (Takara Bio). RT-qPCR using GoTaq qPCR Master Mix (Promega) was performed on a QuantStudio7 Flex (Applied Biosystems). Relative amounts of target mRNAs were calculated using the relative standard curve method and normalized against *actin* (*Glyma.15G050200*) for soybean and *PP2Aa3* (*AT1G13320*) for *Arabidopsis* as a reference gene. Primers used in this study are listed in Supplementary Data 14.

## Measurements of inorganic phosphate contents

Inorganic phosphate (Pi) contents in soybean leaves and *Arabidopsis* seedlings were examined using a molybdate colorimetric assay[73,74]. Frozen powdered soybean leaf samples used for gene expression analysis were also used to analyze Pi content. For *Arabidopsis*, three seedlings from each pot were collected in 2-ml tubes, immediately weighed to determine fresh weight, and frozen in liquid nitrogen. A total of 20–45 mg of each sample was mixed with 1%, v/v, acetic acid (20 μl/mg sample weight), vortexed for 15 min, and incubated at 42 °C for 1 h in a heat block. Following centrifugation at 18,300 × *g* for 5 min at room temperature, the supernatant was used for Pi assay. Reaction solutions containing 140 μl molybdate solution (master mix of 6:1 0.42%, w/v, ammonium molybdate in 1 N $H_2SO_4$ and 10%, w/v, ascorbic acid in water) and 60 μl supernatant were incubated in a 96-well plate at 42 °C for 20 min using a thermal cycler. The absorbance of 100 μl of the reaction solutions at 820 nm was measured using an ARVOX3 plate reader (PerkinElmer, Waltham, MA, USA) or a BioTek Synergy H1

(BioTek Instruments Inc., Winooski, VT, USA). The amount of Pi in the solution was calculated using a calibration curve based on a diluted phosphorus standard solution (FUJIFILM Wako, Osaka, Japan) or $KH_2PO_4$.

## Element analysis

The same soybean leaf samples used for RNA-seq analysis were used for element analysis. Powdered and frozen samples were distributed into 2-ml tubes and then lyophilized using an FDU-2200 freeze dryer. Ten milligrams of the dried samples were immersed in 69% (v/v) $HNO_3$ (Kanto Chemical, Tokyo, Japan) at room temperature for 12 h and subsequently heated to 90 °C for 30 min for complete digestion before being diluted with Milli-Q water (Merck Millipore, Burlington, MA, USA). A NexION 350 S inductively coupled plasma mass spectrometer (PerkinElmer, Waltham, MA, USA) with Syngistix for ICP-MS software ver. 1.0 was used to analyze mineral contents ($^{31}$P, $^{39}$K, $^{44}$Ca, $^{24}$Mg, $^{34}$S, $^{56}$Fe, $^{55}$Mn, $^{11}$B, $^{66}$Zn, $^{63}$Cu, and $^{60}$Ni) in the digested sample. The uptake rate of the solution was 1.3 ml min$^{-1}$ and the data acquisition time was 141 seconds. The reference material, NCS DC73349 (bush branches and leaves), was used to validate the ICP-MS measurement. To determine the nitrogen (N) content, dried samples were ground into a powder. Samples of 1 to 2 mg were weighed using a microbalance (BM-22; A&D Company Limited, Japan) and analyzed for total N content using an NC analyzer (Series II CHNS/O Analyzer 2400; PerkinElmer, Waltham, MA, USA).

## Mild drought stress tests in soybean in a greenhouse using pots

Mild drought stress tests in soybean using pots were conducted in a greenhouse (20 ± 10 °C) at JIRCAS to examine how soil volumetric water content (VWC) affects phosphate starvation response (PSR)-related gene expression in soybean. Soybean seeds were germinated on moistened vermiculite in a temperature-controlled greenhouse, and 5-day-old seedlings were transferred to 350-ml pots filled with A horizon soil collected from the JIRCAS experimental field. Seedlings were grown under well-watered conditions (up to a VWC of 43%) for 5 days in a greenhouse. When primary leaves were fully expanded 10 days after sowing, VWC began to be adjusted to five different levels, WC1 (57%), WC2 (50%), WC3 (43%), WC4 (36%), and WC5 (29%), by measuring the weight of the pot and adding water every Monday, Wednesday, and Friday. The ranges of variation in VWCs during the period from day 7 after sowing to day 38 were 42–57% (WC1), 35–50% (WC2), 30–43% (WC3), 24–36% (WC4), and 20–29% (WC5). The fully expanded first trifoliate leaves of 38-day-old seedlings were collected and immediately frozen in liquid nitrogen. Total RNA from leaves was used for gene expression analysis.

## Mild drought stress tests of soybean in a temperature-controlled growth chamber

Mild drought stress tests of soybean in pots were conducted in a temperature-controlled growth chamber[28] equipped with a $CO_2$ regulator (LH-350S, AMC-$CO_2$-1S; Nippon Medical & Chemical Instruments) under a 14-h-light 25 °C/10-h-dark 23 °C photoperiod to examine how soil water content affects the physiological status and Pi and ABA concentrations. Soybean seeds inoculated with *Bradyrhizobium japonicum* (USDA110) were sown in 300-ml pots filled with an equal volume of vermiculite (grain size: 3–5 mm) and water to a VWC of approximately 40%, and the pots were well-watered (up to a VWC of 60%) with liquid medium [0.1 mM $KH_2PO_4$, 0.4 mM $MgSO_4$, 0.23 μM $H_3BO_3$, 0.035 μM $ZnSO_4$, 0.046 μM $MnCl_2$, 0.001125 μM $CuSO_4$, 0.005 μM $Na_2MoO_4$, 0.25 mM Ca($NO_3$)$_2$, 0.19 μM Na-Fe-EDTA, and 0.25 mM $KNO_3$] for 12 days. The nutrient conditions of the vermiculite and liquid medium were not significantly different from those of the field soil used in this study in terms of P content (Supplementary Data 7). VWC was adjusted to eight different levels, i.e., WC1 (75%), WC2 (67%), WC3 (58%), WC4 (50%), WC5 (42%), WC6 (33%), WC7 (25%),

and WC8 (without water until the end), by weighing the pots and adding water every Monday, Wednesday, and Friday from 12 days after sowing.

The ranges of variation in VWCs during the period from day 17 after sowing to day 20 were as follows: WC1 (52–75%), WC2 (51–67%), WC3 (42–58%), WC4 (34–50%), WC5 (28–42%), WC6 (28–33%), WC7 (23–25%), and WC8 (minimum value:18%). The fully expanded first trifoliate leaves of 20-day-old seedlings were collected and immediately frozen in liquid nitrogen for analyzing the concentrations of Pi and ABA ($n = 4$ for each treatment). To determine aboveground biomass, seedlings were cut at the position of the cotyledon and immediately weighed ($n = 4$ for each treatment). Leaf water content was calculated for each interval as $[(FW) − (DW)/ (FW)] \times 100$, where FW and DW are fresh weight and dry weight, respectively, for any given interval.

### Mild drought stress tests in *Arabidopsis thaliana* in a growth chamber using pots

*Arabidopsis thaliana* L. accession Columbia-0 (Col-0, CS60000) and the *phr1 phl1* double mutant were used in this study. The *phr1 phl1* double mutant was generated by crossing *phr1* (SALK_067629) with *phl1* (SALK_079505), both of which were provided by NASC. Primer sets for confirmation of T-DNA insertion and gene expression in the mutant are shown in Supplementary Data 14. *Arabidopsis* seeds were grown on GM agar plates for 6 to 7 days after stratification[28] with a 16-h-light/8-h-dark cycle ($40 \pm 10$ µmol photons $m^{-2} s^{-1}$). Six-day-old seedlings grown on GM agar plates were transplanted into each pot (7.7-cm diameter) filled with equal amounts of vermiculite (grain size: <1.5 mm) (Midorisangyou, Fukuoka, Japan). Pots were soaked in liquid medium [0.1 mM $KH_2PO_4$, 0.4 mM $MgSO_4$, 0.23 µM $H_3BO_3$, 0.035 µM $ZnSO_4$, 0.046 µM $MnCl_2$, 0.001125 µM $CuSO_4$, 0.005 µM $Na_2MoO_4$, 0.25 mM $Ca(NO_3)_2$, 0.19 µM Na-Fe-EDTA, and 0.25 mM $KNO_3$], and seedlings were grown for 7 days in an environmentally controlled growth room at 21 °C under a 16-h-light/8-h-dark cycle ($70 \pm 20$ µmol photons $m^{-2} s^{-1}$). For 10× Pi liquid nutrient tests, the liquid nutrient medium had 10 times the phosphate content (1.0 mM $KH_2PO_4$). For mild drought tolerance tests, the pots were divided into two groups at 13 days after stratification. Pots of control plants were soaked in water, whereas pots of drought-stressed plants were placed on paper towels to reduce soil moisture (Fig. 5a). For rehydration tests, after 6 days of drought-stress treatment, plants grown in pots that had been soaked in water and left for 1 day were sampled for the rehydration treatment sample. The relationships between soil VWC and water potential were 39.9% (−0.0039 MPa, pF 1.6), 32.0% (−0.0098 MPa, pF 2.0), 27.5% (−0.031 MPa, pF 2.5), 25.0% (−0.098 MPa, pF 3.0), and 9.6% (−0.61 MPa, pF 3.8). To determine the aboveground biomass, the fresh aboveground weights of six seedlings were measured at 1, 3, 6, and 7 days after treatments. The collected seedlings were immediately frozen in liquid nitrogen. Total RNA from leaves was used for RNA-seq analysis or RT-qPCR analysis. Maximum rosette radius was measured using ImageJ software (v.1.51).

### Transcriptome analysis of *Arabidopsis* plants grown in pots

Total RNA was extracted from three *Arabidopsis* seedlings per pot using RNAiso Plus (Takara Bio, Japan) according to the manufacturer's instructions. Three biological replicates per treatment were performed for RNA-Seq analysis. The extracted RNA was used to construct paired-end libraries using a TruSeq Stranded mRNA Sample Prep Kit (Illumina). The libraries were sequenced on an Illumina NovaSeq6000 sequencer with a 151-bp paired-end protocol (Macrogen). Trimmomatic v0.39[64] with the options SLIDINGWINDOW:4:20 and MINLEN:40 was used for trimming low-quality reads and adaptor sequences. Trimmed reads were mapped against the reference sequence of TAIR10 (https://www.arabidopsis.org/) using STAR v2.5.1[65] with the options outSAMstrandField: intronMotif; and outFilterType:

BySJout. The mapped reads and TPM (transcripts per million) of each gene was calculated using featureCounts[75] and TPMCalculator[76], respectively. Differential expression analysis was performed using edgeR package version 3.30.3[77]. The up- and down-regulated DEGs satisfying the following criteria, $|log_2(FC)| \geq 1$, TPM value > 0, and $q < 0.05$, were defined as DEGs between control samples and dry conditions.

Hierarchical clustering of 2,397 genes that were differentially expressed ($|log_2(FC)| \geq 1$, TPM value > 0, $q < 0.05$) in response to mild drought stress in at least one sampling in the RNA-seq experiments was constructed using the 'heatmap.2' function of the gplots package v3.1.3 in R software v3.3.3[67]. Representatives of the different types of PSR genes were selected among up-regulated genes based on a previous report (Supplementary Data 6)[21]. For the ABA-responsive marker gene set (Supplementary Data 12)[40], 193 genes were collected from the top of the gene list of DEGs whose expression was up-regulated at 24 h after ABA treatment[40], the same number as for PSR.

### Inorganic phosphate uptake assay using $H_3^{32}PO_4$

Radiolabeled Pi was used for determining Pi uptake under different soil water conditions in *Arabidopsis*. *Arabidopsis* (Col-0, CS60000) was grown under the same conditions as described in Fig. 6. Before drought treatment, all pots were removed from the nutrient solution and kept on paper towels to remove excess water for 10 min. Pots were soaked in 20 kBq/ml $^{32}$Pi solution ($H_3PO_4$, PerkinElmer, Inc., Japan) in filtrated water for 1 h. After $^{32}$Pi treatment, pots were divided into two groups as described in Fig. 6. The aboveground parts of three seedlings were collected from each pot at 1 and 4 days after drought treatment, and $^{32}$Pi radioactivity was measured using an imaging plate (BAS-IP MS, FUJIFILM, Tokyo Japan) and a Typhoon FLA-7000 image reader (Cytiva, Tokyo, Japan). Signal intensities detected in the region of interest (ROI) were measured using ImageJ software.

### Statistical analysis

Statistical tests in this study, except for the hypergeometric enrichment analysis of PSR marker genes (Fig. 3d, Fig. 5e) conducted using the R function "phyper," were performed in GraphPad Prism 9 (GraphPad Software Inc.; San Diego, CA, USA). An unpaired, two-tailed $t$-test or a one-way analysis of variance (ANOVA) with Tukey's test was used in most experiments. For the randomized block tests in the field, a two-tailed paired samples $t$-test was used.

### Reporting summary

Further information on research design is available in the Nature Portfolio Reporting Summary linked to this article.

## Data availability

RNA-seq data are available from the DNA Data Bank of Japan (www.ddbj.nig.ac.jp/) under accession numbers DRA012279 and DRA014734. Source data are provided with this paper.

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

## Acknowledgements

We thank S. Shu, M. Toyoshima, K. Ozawa, N. Komatsu, H. Ishiyama, K. Matayoshi, K. Kawamura, N. Hisatomi, Y. Saito, Y. Shirai, Y. Nakamura, Y. Nonoue, Y. Takiguchi, Y. Masamura, A. Aoyama, K. Mitadera, I. Gejima, H. Ohwada, N. Tayama, K. Watanabe, T. Sekine, S. Nakamura, K. Yoshihara, and S. Takasugi for technical assistance, and J. Kikuchi, Y. Tsuboi, K. Yoshida, T. Fujiwara, T. Taji, K. Nakashima, M. Fujita, T. Ogata, Y. Murata, T. Kashiwa, O. Rollano-Peñaloza, J. Quezada, and S. Neyrot for discussions. This work was supported by Grants-in-Aid for Scientific Research (KAKENHI) from the Japan Society for the Promotion of Science (JSPS) (Grant Nos. JP18K05379 to Y.N.; JP21H02158 to Y.N., Y.F.; JP16K07412, JP24510312 to Y.F.), the Cabinet Office, Government of Japan, Moonshot Research and Development Program for Agriculture, Forestry and Fisheries (funding agency: Bio-oriented Technology Research Advancement Institution; Grant No. JPJ009237), the Japan International Cooperation Agency (JICA) for the Science and Technology Research Partnership for Sustainable Development (SATREPS; Grant No. JPMJSA1907), and the Ministry of Agriculture, Forestry and Fisheries (MAFF) of Japan.

## Author contributions

Y.N. and Y.F. conceived and designed the study and wrote the paper; Y.N. performed most of the experiments and analyzed the data; K.I. performed the soil and chemical analyses; Y.N., K.I., K.F., and Y.F. performed the field experiments, with support from T.O. and Y.K.; R.S. performed the analysis using radioisotopes; Y.T., M.K., and H.S. performed quantitative analyses of abscisic acid; N.K., K.T., and R.S. performed elemental analyses; J.B. assisted Y.N.'s analyses; Y.N., N.M., Y.K., E.O-T., M.I., and Y.Y. performed the transcriptome analysis. All authors discussed and commented on the manuscript.

## Competing interests

The authors declare no competing interests.
