## [Peer Review File · Nature Communications]

Phosphate starvation response precedes abscisic acid response under progressive mild drought in plantsReviewer #1 (Remarks to the Author):

Nagatoshi and colleagues reported an interesting observation that phosphate deficiency signaling (PSR) proceeds ABA response to drought in plants. The authors presented interesting and converging results obtained from plants grown in the field with several repeats and in the Lab conditions. The results are presented in a nice manner, and the text is easy to read and follows logic.

That being said, I have a few important concerns with the interpretation of the results, that I believe are needed to be addressed to make a significant contribution in the field.

1- The link between Pi and drought stress is based on gene expression changes. While this is a good indicator but it remains the first indicator. To me, the most important result may come from the fact that plants accumulate less Pi when grown on ridges compared to flat (Figure 3g). Worth noting that Pi deficiency response really induces these markers to a greater level, and not only 2 to 4 levels. Nevertheless, here we can see that in the third trifoliate leaf there is no induction of PHR1 (the master regulator) while still the other markers such SPX etc are induced.

Now, my concern is that the total inorganic phosphate (Pi) measurement isn't at the level of detail that satisfies the readers' curiosities. i) NMR can tell which pool of Pi is affected. This is an easy and highly informative experiment. ii) More importantly, in the Pi signaling research, a particular focus should be to determine the changes in inositol phosphate (InsPs) concentration, as we know that the decrease of a class of inositol phosphate triggers PSR via the dissociation between PHR1 and SPX proteins and that a link between InsP and drought is established in the literature. These details are expected from the readers more than a simple Pi measurement via a colorimetric method.

2- The combination between Soybean and Arabidopsis is really great and shows consistency in the obtained results. More importantly, the use of Arabidopsis offers the authors the possibility to test using mutants in PSR genes the response to drought. These are missing data to reinforce/provide the first answer to WHY PSR and Pi homeostasis is induced before ABA/drought responses. Providing elements of response to these questions is expected, to fully appreciate the authors' observation of gene expression patterns.

3- The authors are talking about a reduction in Pi uptake. I could see data supporting this statement in line 227. Pi uptake is different from Pi accumulation in roots and needs to be addressed using radiolabeled P.

4- Finally, the table provided by the authors on gene differentially expressed revealed that transporter of other nutrient is affected as indicated by authors line 231, it is interesting to perform an easy ionome analysis to describe to the readers what the other nutrient affected under different treatment, and/or what is specific for Pi. This will clarify the situation to the hypothesis that plant accumulates less nutrient because they grow less.

Reviewer #2 (Remarks to the Author):

The authors use ridges in field to have the soil dry out a bit faster than without ridges along with transcriptome data to propose that in plants subjected to mild reduction in soil water content the phosphate starvation response is up-regulated before ABA-regulated changes in gene expression. Overall, I think it is entirely plausible that the plant response to mild water limitation induces a different set of genes than the response to severe water limitation and dehydration and that ABA-responses are mostly associated with the latter. There is actually lots of evidence in the literature to support such a conclusion, almost none of which is cited in this study. They find that the plants grown on ridges have reduced tissue Pi content. However, they did not compare ABA levels between the two treatments. They also do not do what would seem to be the obvious experiment to test whether a higher level of soil phosphate can alleviate the growth and transcriptional phenotypes they find to be associated with mild soil drying.

To show that the effects they observe are not specific to one type of soil, the authors also do

experiments in Arabidopsis plants grown in drying vermiculite. They seem to get a similar transcriptional response as the soil experiments but do not measure plant tissue phosphate levels or ABA levels and do not report water potentials of the vermiculite to allow comparison of the soil treatments. Even though they propose that the phosphate starvation response precedes the ABA response, many experiments that could further support and validate such a proposition are left undone. In particular, they do not test whether the plant growth response to mild drying is altered in different phosphate treatments and do not test whether mutants with impaired phosphate sensing or uptake response differently to mild drying. Such experiments, particularly the testing of phosphate sensing/uptake mutants is needed because without such experiments the main conclusion is based entirely on gene ontology analysis of their transcriptome data. Gene ontology annotations are often incomplete and hard to interpret, particularly for stress treatments. Some of the genes included in the phosphate starvation annotation may also in fact be regulated in response to drought independently of phosphate, just that this has not been tested or reported in the studies that were used as the basis for the GO annotations.

Because of these limitations, I regard their results as potentially very interesting but not conclusively demonstrated. There is also an unresolved question of cause and effect: does the water limitation (reduced water uptake) and reduced growth cause the phosphate starvation response to be induced, or is reduced phosphate uptake the cause of the reduced growth that occurs during mild water deficit. For now all one can say is that these things appear to be correlated. Determining which one is the important factor causing reduced plant growth would greatly increase the impact of this study.

Specific comments:

1. Line 117-130: This paragraph is not an accurate summary of the state of drought research or the methods used. There are laboratory methods have been used quite successfully to impose consistent moderate severity drought stress on seedlings (PEG-agar plates are particularly useful for this purpose). In growth chamber experiments, controlled watering experiments to impose moderate severity water limitation over an extended time are increasingly common. Rainout shelters can be used to control drought severity in field experiments. None of these are mentioned in this paragraph and thus the authors give an incomplete, at best, picture of drought research.
2. Water potentials should be expressed in MPa throughout as this is the standard unit to use, not kPa.
3. At several places in the results, the authors mention reduced transpiration of plants grown on ridges; however, they do not report any actual measurements of transpiration. This should be corrected in the text.
4. Figure 2C does not specify the units used (grams presumably??), same problem in some of the supplemental figures.
5. In Figure 4B it is not at all clear that the leaf temperature increases in proportion to the level of soil drying as stated in the text. Partially this is because the odd and unclear false color scheme used in Fig 4B. In addition, the authors need to provide quantitative data to support such statement. The authors should also report soil water potentials for each of the four treatments. And, why label the treatments with the very confusing Greek letters? Measure the water potential of each treatment and then plot all of the data versus to soil water potential to see what the actual relationship of phenotype to soil water potential is.
6. The authors do not supply water potential measurements for the vermiculite drying treatments, thus there is no way to compare the stress severity in these treatments to the severity in soil treatments as the relationship between water content and water potential differs for different soil type (see Dowd et al., 2021 Plant Cell and Environment for example of this).
7. The authors do not include any information on the phosphate content of the vermiculite in the drying treatment and how it compares to the soil experiments.
8. Line 320-330: much relevant literature about what induced ABA accumulation is not cited here. The authors should also do more to test these hypotheses, for example by measuring leaf water content.
9. Line 339: this is why your analysis should be based on water potential, not soil water content.
10. Line 350-354: the authors over-hype the usefulness of ridges, which are not likely to work in areas with different rainfall frequencies or soil type.
11. There are other transcriptome data sets of plants subjected to moderate low water potential stress (for example, the Des Marias et al study that is cited, as well as others). The authors should compare their transcriptome data to these other data sets to see how much overlap there is. Has this enrichment of phosphate starvation response genes occurred in other studies and been

overlooked, or interpreted in a different manner compared to here? If it was not observed in other studies, what are possible reasons for this?

Response to the Reviewers

RESPONSE 1:

We thank the editor and reviewers for their valuable comments and suggestions to improve our manuscript. All the changes and additions made in response to the editor's and reviewers' comments are in blue font in the revised manuscript. We followed all the instructions here and are prepared to resubmit our manuscript files.

The other points have been altered as follows:

Lines 1–2: As we have added new data, we have updated the title of our revised manuscript to “Phosphate starvation response precedes abscisic acid response under progressive mild drought in plants”.

Lines 4–7, 14–19, 28, 38–41: We have added new authors who were involved in obtaining the new data and have revised the order of authors according to changes in their relative contributions to the paper.

Lines 52–55, 137–140: The abstract and introduction have been updated to include the new data.

Line 76: Word usage has been revised throughout the text to reflect changes resulting from the new data.

Lines 295–297; Fig. 1d, f; Fig. 2, Fig. 3g; Fig. 5; Supplementary Fig. 6; Supplementary Fig. 7; Supplementary Fig. 8: When adding new data, statistical analysis methods were reviewed for existing data to facilitate comparison with existing data, and units were changed for existing data as appropriate to facilitate comparison.

Line 429: New information has been added to the text to reflect the addition of new data.

Lines 77, 269, 279–281, 301, 312, 378, 379–381, 429, 484, 511, 528, 531–556, 621, 642, 648–655, 657–658, 663–664, 668–672, 679–681, 683–694, 700, 711–734, 737–742, 745–750, 760–761, 781, 787–796, 800, 802–803: We have also altered the sentences and added new descriptions to the Materials and Methods section to accommodate the newly added data and have corrected text in the previous version of the manuscript as appropriate.

Lines 810–823: Acknowledgments have been updated to include individuals who provided support for work that yielded the new data.

Lines 835–836, 840, 848–849, 850, 853, 871–873, 876–877, 883, 886–888, 890–891, 902, 904–907, 909, 911, 929–930, 936–937, 945–947, 954–957, 967, 971–974, 997–999,

1003: Errors in the references cited in the previous version of the manuscript have been corrected.

Lines 897–899, 939–940, 951, 969–970, 988–996: When creating the revised version with the new data, new references were also added.

En dashes, hyphens, and minus signs were corrected as needed throughout the revised manuscript. As new data were added, the numbers of Figures, Supplementary Figures, and Supplementary Data have been updated accordingly.

COMMENT 2:

REVIEWER COMMENTS

Reviewer #1 (Remarks to the Author):

Nagatoshi and colleagues reported an interesting observation that phosphate deficiency signaling (PSR) proceeds ABA response to drought in plants. The authors presented interesting and converging results obtained from plants grown in the field with several repeats and in the Lab conditions. The results are presented in a nice manner, and the text is easy to read and follows logic.

That being said, I have a few important concerns with the interpretation of the results, that I believe are needed to be addressed to make a significant contribution in the field.

RESPONSE 2:

Thank you for your positive evaluation of our manuscript. We have addressed all of your comments in a point-by-point manner below.

COMMENT 3:

1- The link between Pi and drought stress is based on gene expression changes. While this is a good indicator but it remains the first indicator. To me, the most important result may come from the fact that plants accumulate less Pi when grown on ridges compared to flat (Figure 3g). Worth noting that Pi deficiency response really induces these markers to a greater level, and not only 2 to 4 levels. Nevertheless, here we can see that in the third trifoliolate leaf there is no induction of PHR1 (the master regulator) while still the other markers such SPX etc are induced.

RESPONSE 3:

Thank you for pointing this out. The expression levels of all PSR genes examined were, to varying degrees, lower in the third trifoliolate leaves than in the first and second trifoliolate leaves of soybean plants grown on ridges (Fig. 3f), and the expression levels of PSR genes tended to be inversely correlated with the Pi contents in the leaves (Fig. 3f, g). In response to this comment, we have added the following sentences in the Results section (Lines 228–232):

“Notably, Pi concentrations in the third trifoliolate leaves of soybean plants grown on ridges were higher than those in the first and second trifoliolate leaves, and Pi concentrations

tended to be inversely related to varying degrees with the expression levels of PSR marker genes (Fig. 3f, g). These results suggest that mild drought stress reduced Pi concentrations in the leaves of field-grown soybean plants, resulting in PSR induction.”

COMMENT 4:

Now, my concern is that the total inorganic phosphate (Pi) measurement isn't at the level of detail that satisfies the readers' curiosities. i) NMR can tell which pool of Pi is affected. This is an easy and highly informative experiment. ii) More importantly, in the Pi signaling research, a particular focus should be to determine the changes in inositol phosphate (InsPs) concentration, as we know that the decrease of a class of inositol phosphate triggers PSR via the dissociation between PHR1 and SPX proteins and that a link between InsP and drought is established in the literature. These details are expected from the readers more than a simple Pi measurement via a colorimetric method.

RESPONSE 4:

Thank you for your interesting suggestions. i) For NMR, we consulted with NMR experts at RIKEN and proceeded with the experiment. However, we found that there were no similar reference studies and many technical problems, which made it difficult to present the NMR results in this paper. ii) For InsP analysis, we consulted with InsP researchers at the University of Tokyo and other institutions but found that the quantification of InsP is technically challenging and a topic that only a few groups in the world have tackled. In addition, RI-labeled InsP is essential for its measurement. However, there are no examples of reference studies that have quantified InsP in a drought experimental system involving fields and pots, such as the system used in our study, and considering the technical and equipment barriers, it would be difficult to include these data in this paper. Therefore, we hope to address these problems in the future.

COMMENT 5:

2- The combination between Soybean and Arabidopsis is really great and shows consistency in the obtained results. More importantly, the use of Arabidopsis offers the authors the possibility to test using mutants in PSR genes the response to drought. These are missing data to reinforce/provide the first answer to WHY PSR and Pi homeostasis is induced before ABA/drought responses. Providing elements of response to these questions is expected, to fully appreciate the authors' observation of gene expression patterns.

RESPONSE 5:

Thank you so much for your positive evaluation of our work combining soybean analyses in the field and lab with *Arabidopsis* analyses in the lab. In response to the reviewer's suggestions, as we described in RESPONSE 12, we have performed experiments to induce mild drought using *Arabidopsis phr1phl1* double mutant plants, which have a deficient PSR. Even under mild drought stress, where control plants can continue to grow, the growth of the *phr1phl1* mutant almost stopped, suggesting that PSR plays a crucial role in plant growth under mild drought. We have added new data (Figure 6; Supplementary Fig. 16) and the following paragraph in the Results section (Lines 342–361):

" **PSR plays a crucial role in plant growth under mild drought.** Even when grown with excess Pi (10× Pi: 10 times the amount of Pi), mild drought significantly reduced Pi concentrations compared with those in control plants (Supplementary Fig. 14e), but failed to induce PSR (Supplementary Fig. 14i). Excess promoted plant growth in both control and mild drought-treated plants (Supplementary Fig. 14f–h). These observations suggest that the PSR induced when Pi concentrations in plants fall below a certain threshold affects plant growth under mild drought. Further, to examine the role of PSR under mild drought in planta, we induced mild drought stress in *Arabidopsis phr1 phl1* double mutant plants (Fig. 6; Supplementary Fig. 16), which are deficient in PSR^{41, 42}. Although there was variation, Pi concentrations were lower under mild drought treatment than under control treatment in both the *phr1 phl1* double mutant and wild-type plants (Fig. 6c, d). Interestingly, when wild-type plants were rewatered, Pi concentrations increased rapidly, but no significant increase in Pi concentration was observed in the *phr1 phl1* double mutant (Fig. 6c, d), indicating that the PSR is responsible for the rapid increase in Pi concentration during rewatering. To evaluate the effects of the PSR on growth, we measured the aboveground biomass of seedlings and maximum rosette radius after 1, 3, 6, and 7 days of drought stress treatment (Fig. 6e, f). Both the *phr1 phl1* mutant and wild-type plants grew to varying degrees under control conditions; however, under mild drought conditions, the wild-type plants continued to grow, albeit poorly, while the *phr1 phl1* mutant plants showed no significant growth (Fig. 6e–h). These data indicate that PSR fulfills a crucial role in plant growth under progressive mild drought."

COMMENT 6:

3- The authors are talking about a reduction in Pi uptake. I could see data supporting this statement in line 227. Pi uptake is different from Pi accumulation in roots and needs to be addressed using radiolabeled P.

RESPONSE 6:

Thank you for your helpful suggestion. As described in RESPONSE 13, we used ³²Pi to determine if Pi uptake is affected by mild drought. We found that mild drought reduces Pi uptake from the roots, which results in reduced Pi concentrations in the aboveground organs of *Arabidopsis* seedlings. We have added data (Supplementary Fig. 15) and the following paragraph (Lines 325–331):

Lines 325–331: **Reduced Pi uptake rather than reduced growth causes PSR.** It remains unclear whether mild drought reduces the Pi concentration in plants by reducing Pi uptake or by inhibiting growth in response to reduced soil moisture. To test whether the reduction in Pi is a consequence of reduced Pi uptake, we performed uptake experiments with radiolabeled ³²Pi (Supplementary Fig. 15a). ³²Pi signals were detected in the aboveground parts of control plants on day 4 but not in the aboveground parts of drought-treated plants (Supplementary Fig. 15b, c), indicating that mild drought inhibits Pi uptake in plants, reducing Pi in plants.

COMMENT 7:

4- Finally, the table provided by the authors on gene differentially expressed revealed that transporter of other nutrient is affected as indicated by authors line 231, it is interesting to perform an easy ionome analysis to describe to the readers what the other nutrient affected under different treatment, and/or what is specific for Pi. This will clarify the situation to the hypothesis that plant accumulates less nutrient because they grow less.

RESPONSE 7:

Thank you for your useful suggestion. We performed element analysis on the leaf samples used for RNA-seq analysis. Among the major macronutrients, only P was markedly reduced in the leaves of soybean plants grown on ridges compared to those of soybean plants grown on flats. We have added the data (Supplementary Fig. 12) and the following sentences (Lines 233–244, 248–250) and altered one sentence (Lines 250–251):

Lines 233–244: We performed elemental analysis using inductively coupled plasma mass spectrometry (ICP-MS) and ICP-optical emission spectrometry (ICP-OES) to determine if other inorganic nutrients were also affected by mild drought (Supplementary Fig. 12). Of the three primary macronutrients, N, P, and K, which are essential for plant growth, only P was markedly lower in the leaves of soybean plants grown on ridges compared with those of soybean plants grown on flats (Supplementary Fig. 12a). Among the secondary macronutrients essential for plant growth, magnesium (Mg) and sulfur (S) contents in the leaves of soybean plants grown on ridges were slightly lower than those of soybean plants grown on flats, but no drastic changes were observed (Supplementary Fig. 12b). Among the micronutrients essential for plant growth, boron (B), zinc (Zn), and nickel (Ni) were significantly higher, whereas copper (Cu) was significantly lower in leaves of soybean plants grown on ridges compared with those of plants grown on flats (Supplementary Fig. 12c).

Lines 248–250: This is consistent with our observation that mild drought conditions reduced P by 47.3% ($\pm 7.8\%$), whereas nitrogen was reduced by only 11.6% ($\pm 5.9\%$) (Supplementary Fig. 12a).

Lines 250–251: These findings support the notion that mild drought stress reduces levels of Pi, among nutrient elements, and induces PSR in plants in the field.

COMMENT 8:

Reviewer #2 (Remarks to the Author):

The authors use ridges in field to have the soil dry out a bit faster than without ridges along with transcriptome data to propose that in plants subjected to mild reduction in soil water content the phosphate starvation response is up-regulated before ABA-regulated changes in gene expression. Overall, I think it is entirely plausible that the plant response to mild water limitation induces a different set of genes than the response to severe water limitation and dehydration and that ABA-responses are mostly associated with the latter. There is actually lots of evidence in the literature to support such a conclusion, almost none of which is cited in this study.

RESPONSE 8:

Thank you for your comment. We had mentioned your point in the Introduction, along with the relevant references, but in response to your suggestion, we have added one more important reference (Line 85), taking into account the limitation on the number of references.

COMMENT 9:

They find that the plants grown on ridges have reduced tissue Pi content. However, they did not compare ABA levels between the two treatments.

RESPONSE 9:

Thank you for your interesting comments. In response to this comment, we measured ABA levels in the leaf samples used for RNA-seq analysis. We found that there was no clear difference in ABA contents of soybean leaves of plants grown on the flats and ridges. We have added these data (Supplementary Fig. 10) and the following sentence (Lines 212–214):

"This result is consistent with the fact that there was no clear difference in ABA content between the leaves of soybean plants grown on flats and those of plants grown on ridges (Supplementary Fig. 10)."

COMMENT 10:

They also do not do what would seem to be the obvious experiment to test whether a higher level of soil phosphate can alleviate the growth and transcriptional phenotypes they find to be associated with mild soil drying.

RESPONSE 10:

Thank you for your interesting point. In response, as we described in RESPONSE 12, we conducted the experiments under higher levels of soil phosphate. We have added the data (Supplementary Fig. 14) with the following sentences (Lines 342–348):

"Even when grown with excess Pi (10× Pi: 10 times the amount of Pi), mild drought significantly reduced Pi concentrations compared with those in control plants (Supplementary Fig. 14e), but failed to induce PSR (Supplementary Fig. 14i). Excess promoted plant growth in both control and mild drought-treated plants (Supplementary Fig. 14f–h). These observations suggest that the PSR induced when Pi concentrations in plants fall below a certain threshold affects plant growth under mild drought."

COMMENT 11:

To show that the effects they observe are not specific to one type of soil, the authors also do experiments in Arabidopsis plants grown in drying vermiculite. They seem to get a similar transcriptional response as the soil experiments but do not measure plant tissue phosphate levels or ABA levels and do not report water potentials of the vermiculite to allow comparison of the soil treatments. Even though they propose that the phosphate

starvation response precedes the ABA response, many experiments that could further support and validate such a proposition are left undone.

RESPONSE 11:

Thank you for your valuable suggestions. In response, as described in RESPONSE 13, we have quantified ABA and Pi over time in mild drought tests using *Arabidopsis thaliana*. We have added these data (Supplementary Fig. 14) and the following sentences (Lines 317–323, 332–340):

Lines 317–323: These data show that mild drought stress induces expression of PSR genes before inducing ABA-responsive genes (Fig. 5), consistent with the observation that the Pi concentration gradually decreases during mild drought (Supplementary Fig. 14d), whereas the ABA content increases as drought exceeds a certain level (Supplementary Fig. 14c). Together with the results of mild stress trials in soybean plants grown in pots and the field, these results demonstrate that the PSR induced by Pi reduction is initiated before ABA response in plants grown under progressive mild drought conditions.

Lines 332–340: We examined whether the reduction in Pi caused by mild drought was responsible for the reduced growth. Rehydration after 6 days of drought treatment rapidly increased Pi concentration (Supplementary Fig. 14d) and rapidly decreased PSR gene expression (Fig. 5f, g; Supplementary Fig. 14i), but there was no clear difference in aboveground biomass between plants after 7 days of drought treatment and plants rehydrated for 1 day after 6 days of drought treatment (Fig. 5c; Supplementary Fig. 14f). These observations indicate that the reduced Pi due to mild drought is involved in response to soil moisture conditions rather than plant growth. Taken together, these findings support the notion that mild drought inhibits Pi uptake, resulting in reduced Pi and thus inducing PSR in plants.

We have also added the water potential values (Lines 168, 171, 206, 263–265, 293–294, 754–757; Figs. 1, 4–6; Supplementary Figs. 7–9, 13, 14), as we described in RESPONSE 18, RESPONSE 19, and RESPONSE 22.

COMMENT 12:

In particular, they do not test whether the plant growth response to mild drying is altered in different phosphate treatments and do not test whether mutants with impaired phosphate sensing or uptake response differently to mild drying. Such experiments, particularly the testing of phosphate sensing/uptake mutants is needed because without such experiments the main conclusion is based entirely on gene ontology analysis of their transcriptome data. Gene ontology annotations are often incomplete and hard to interpret, particularly for stress treatments. Some of the genes included in the phosphate starvation annotation may also in fact be regulated in response to drought independently of phosphate, just that this has not been tested or reported in the studies that were used as the basis for the GO annotations.

RESPONSE 12:

Thank you for your useful suggestions. As we described in RESPONSE 10, we conducted the experiments under higher levels of soil phosphate. We have added these data (Supplementary Fig. 14) and the following sentences (Lines 342–348):

"Even when grown with excess Pi (10× Pi: 10 times the amount of Pi), mild drought significantly reduced Pi concentrations compared with those in control plants (Supplementary Fig. 14e), but failed to induce PSR (Supplementary Fig. 14i). Excess promoted plant growth in both control and mild drought-treated plants (Supplementary Fig. 14f–h). These observations suggest that the PSR induced when Pi concentrations in plants fall below a certain threshold affects plant growth under mild drought."

In addition, as we described in RESPONSE 5, we have performed experiments to induce mild drought using *Arabidopsis phr1phl1* double mutant plants, which have a deficient PSR. Even under mild drought stress, where control plants continued to grow, the growth of the *phr1phl1* mutant almost stopped, suggesting that PSR plays a crucial role in plant growth under mild drought. We have added new data (Fig. 6; Supplementary Fig. 16) and the following paragraph in the Results section (Lines 342–361):

"PSR plays a crucial role in plant growth under mild drought. Even when grown with excess Pi (10× Pi: 10 times the amount of Pi), mild drought significantly reduced Pi concentrations compared with those in control plants (Supplementary Fig. 14e), but failed to induce PSR (Supplementary Fig. 14i). Excess promoted plant growth in both control and mild drought-treated plants (Supplementary Fig. 14f–h). These observations suggest that the PSR induced when Pi concentrations in plants fall below a certain threshold affects plant growth under mild drought. Further, to examine the role of PSR under mild drought in planta, we induced mild drought stress in *Arabidopsis phr1 phl1* double mutant plants (Fig. 6; Supplementary Fig. 16), which are deficient in PSR^{41, 42}. Although there was variation, Pi concentrations were lower under mild drought treatment than under control treatment in both the *phr1 phl1* double mutant and wild-type plants (Fig. 6c, d). Interestingly, when wild-type plants were rewatered, Pi concentrations increased rapidly, but no significant increase in Pi concentration was observed in the *phr1 phl1* double mutant (Fig. 6c, d), indicating that the PSR is responsible for the rapid increase in Pi concentration during rewatering. To evaluate the effects of the PSR on growth, we measured the aboveground biomass of seedlings and maximum rosette radius after 1, 3, 6, and 7 days of drought stress treatment (Fig. 6e, f). Both the *phr1 phl1* mutant and wild-type plants grew to varying degrees under control conditions; however, under mild drought conditions, the wild-type plants continued to grow, albeit poorly, while the *phr1 phl1* mutant plants showed no significant growth (Fig. 6e–h). These data indicate that PSR fulfills a crucial role in plant growth under progressive mild drought."

In response to this suggestion, we have shown that the transcriptome data (Figs. 3, 5) strongly correlated with changes in Pi concentration and ABA levels, based on the newly added results of Pi and ABA quantification, as well as results obtained under different Pi conditions (Fig. 6; Supplementary Figs. 10, 13, 14). We have added the following sentences (Lines 212–214, 269–275, 317–323).

Lines 212–214: This result is consistent with the fact that there was no clear difference in ABA content between the leaves of soybean plants grown on flats and those of plants grown on ridges (Supplementary Fig. 10).

Lines 269–275: This result was supported by observations in pot tests using vermiculite (Supplementary Data 7), which is commonly used as an artificial soil for experiments, and liquid nutrients with 0.1 mM KH_2PO_4 . With increasing severity of mild drought (Supplementary Fig. 13a, b), aboveground biomass and leaf water content gradually decreased (Supplementary Fig. 13c, d), whereas the Pi concentration initially decreased but later increased (Supplementary Fig. 13e). By contrast, the ABA content increased as drought exceeded a certain level (Supplementary Fig. 13f).

Lines 317–323: These data show that mild drought stress induces expression of PSR genes before inducing ABA-responsive genes (Fig. 5), consistent with the observation that the Pi concentration gradually decreases during mild drought (Supplementary Fig. 14d), whereas the ABA content increases as drought exceeds a certain level (Supplementary Fig. 14c). Together with the results of mild stress trials in soybean plants grown in pots and the field, these results demonstrate that the PSR induced by Pi reduction is initiated before ABA response in plants grown under progressive mild drought conditions.

COMMENT 13:

Because of these limitations, I regard their results as potentially very interesting but not conclusively demonstrated. There is also an unresolved question of cause and effect: does the water limitation (reduced water uptake) and reduced growth cause the phosphate starvation response to be induced, or is reduced phosphate uptake the cause of the reduced growth that occurs during mild water deficit. For now all one can say is that these things appear to be correlated. Determining which one is the important factor causing reduced plant growth would greatly increase the impact of this study.

RESPONSE 13:

Thank you so much for your interest in our results and for raising this important point. We have conducted uptake experiments with ^{32}P (Supplementary Fig. 15) to answer this question, as described in RESPONSE 6, and quantification of Pi during mild drought (Supplementary Fig. 14), as described in RESPONSE 11. These experiments led to the conclusion that reduced Pi uptake, rather than reduced growth, causes PSR. Based on our new findings, we have added the following paragraph in the Results section (Lines 325–340).

Lines 325–340: **Reduced Pi uptake rather than reduced growth causes PSR.** It remains unclear whether mild drought reduces the Pi concentration in plants by reducing Pi uptake or by inhibiting growth in response to reduced soil moisture. To test whether the reduction in Pi is a consequence of reduced Pi uptake, we performed uptake experiments with radiolabeled ^{32}Pi (Supplementary Fig. 15a). ^{32}Pi signals were detected in the aboveground parts of control plants on day 4 but not in the aboveground parts of

drought-treated plants (Supplementary Fig. 15b, c), indicating that mild drought inhibits Pi uptake in plants, reducing Pi in plants.

We examined whether the reduction in Pi caused by mild drought was responsible for the reduced growth. Rehydration after 6 days of drought treatment rapidly increased Pi concentration (Supplementary Fig. 14d) and rapidly decreased PSR gene expression (Fig. 5f, g; Supplementary Fig. 14i), but there was no clear difference in aboveground biomass between plants after 7 days of drought treatment and plants rehydrated for 1 day after 6 days of drought treatment (Fig. 5c; Supplementary Fig. 14f). These observations indicate that the reduced Pi due to mild drought is involved in response to soil moisture conditions rather than plant growth. Taken together, these findings support the notion that mild drought inhibits Pi uptake, resulting in reduced Pi and thus inducing PSR in plants.

COMMENT 14:

Specific comments:

1. Line 117-130: This paragraph is not an accurate summary of the state of drought research or the methods used. There are laboratory methods have been used quite successfully to impose consistent moderate severity drought stress on seedlings (PEG-agar plates are particularly useful for this purpose). In growth chamber experiments, controlled watering experiments to impose moderate severity water limitation over an extended time are increasingly common. Rainout shelters can be used to control drought severity in field experiments. None of these are mentioned in this paragraph and thus the authors give an incomplete, at best, picture of drought research.

RESPONSE 14:

Thank you for your comments. Based on these comments, we have amended the sentences as follows (Lines 119–129):

"For example, in indoor growth chambers, mild drought stress tests have been carried out using agar medium containing polyethylene glycol to induce osmotic stress in seedlings, or soil in small pots to provide mild drought over an extended period of time by manually or mechanically controlling and monitoring soil moisture^{30, 31}. In greenhouses, drought stress tests have been performed to vary irrigation conditions for plants grown in pots, whereas in the field, the tests have been done in experimental plots with different irrigation conditions and rainout shelters to keep rain out of the test plots³². While it is relatively easy to control the growing environment indoors, it is not easy to mimic the complex drought conditions that occur in the field. Field tests do not have the various constraints that are present in pot tests, but complex changes in the natural environment make it challenging to control environmental conditions such as drought."

COMMENT 15:

2. Water potentials should be expressed in MPa throughout as this is the standard unit to use, not kPa.

RESPONSE 15:

Thank you for your comment. As suggested, the unit was changed from kPa to MPa throughout.

COMMENT 16:

3. At several places in the results, the authors mention reduced transpiration of plants grown on ridges; however, they do not report any actual measurements of transpiration. This should be corrected in the text.

RESPONSE 16:

Thank you for pointing this out. As suggested, we have deleted the misleading description (Lines 179–180).

COMMENT 17:

4. Figure 2C does not specify the units used (grams presumably??), same problem in some of the supplemental figures.

RESPONSE 17:

Thank you for pointing this out. We have corrected these errors (Fig. 2; Supplementary Fig. 8).

COMMENT 18:

5. In Figure 4B it is not at all clear that the leaf temperature increases in proportion to the level of soil drying as stated in the text. Partially this is because the odd and unclear false color scheme used in Fig 4B. In addition, the authors need to provide quantitative data to support such statement. The authors should also report soil water potentials for each of the four treatments. And, why label the treatments with the very confusing Greek letters? Measure the water potential of each treatment and then plot all of the data versus to soil water potential to see what the actual relationship of phenotype to soil water potential is.

RESPONSE 18:

Thank you for all your comments. To make the temperature distribution quantitatively easier to understand, the color scheme in Figure 4b was changed and the figure was enlarged so that the pots appear larger. The reason the labels for each drought treatment were originally in Greek letters was because it was confusing to mark them with a single letter of the alphabet.

In response to the reviewer's comments, we have changed the notation to a combination of two alphabetic letters and numbers, such as WC1–5 (Lines 257–258, 263–265, 704, 706–707; Fig. 4; Supplementary Fig. 13). WC indicates the water condition.

As we described in RESPONSE 11, RESPONSE 19, and RESPONSE 22, this study was originally designed on a soil moisture content basis, and in light of the fact that a certain number of researchers are unfamiliar with the use of water potential, the values of moisture content and water potential are listed together for easy understanding by a wider range of readers (Lines 168, 171, 206, 263–265, 293–294, 754–757; Figs. 1, 4–6; Supplementary Figs. 7–9, 13, 14).

COMMENT 19:

6. The authors do not supply water potential measurements for the vermiculite drying treatments, thus there is no way to compare the stress severity in these treatments to the severity in soil treatments as the relationship between water content and water potential differs for different soil type (see Dowd et al., 2021 Plant Cell and Environment for example of this).

RESPONSE 19:

Thank you for your comment. Based on this comment, we have added the water potential values (Lines 168, 171, 206, 263–265, 293–294, 754–757; Figs. 1, 4–6; Supplementary Figs. 7–9, 13, 14), as described in RESPONSE 11, RESPONSE 18, and RESPONSE 22.

COMMENT 20:

7. The authors do not include any information on the phosphate content of the vermiculite in the drying treatment and how it compares to the soil experiments.

RESPONSE 20:

Thank you for pointing this out. Based on your input, we have added the data on the phosphate content of vermiculite (Supplementary Data 7). To show that the same amount of phosphoric acid was given to plants exposed to mild drought stress and control plants and that the amount of phosphorus was not significantly different between the soil and vermiculite experiments, we have added the following sentences (Lines 284–287, 720–722):

Lines 284–287: Pots containing control plants were soaked in water without nutrients, while pots containing drought-stressed plants were placed on paper towels to reduce soil moisture (Fig. 5a), meaning that the vermiculite in pots of both treatments contained the same amount of nutrients, including phosphate.

Lines 720–722: Nutrient conditions of the vermiculite and liquid medium were not significantly different from those of the field soil used in this study in terms of P content (Supplementary Data 7).

COMMENT 21:

8. Line 320-330: much relevant literature about what induced ABA accumulation is not cited here. The authors should also do more to test these hypotheses, for example by measuring leaf water content.

RESPONSE 21:

Thank you for your comment. Considering the limitations in the number of references that can be cited, we have cited the literature on the induction of ABA accumulation in plants subjected to drought stress (Lines 387–399). To test the hypothesis posed in the Discussion of the previous version of our manuscript, we have quantified Pi and ABA (Supplementary Figs. 10, 13, and 14), performed element analysis (Supplementary Fig. 12), conducted drought stress tests using high P content soil (Supplementary Fig. 14),

performed uptake analysis using ^{32}P i (Supplementary Fig. 15), analyzed PSR-deficient mutant plants (Fig. 6; Supplementary Fig. 16), and added more data. Based on these data, we have substantially revised the description in this section (Lines 387–399) as follows:

" We also demonstrate that PSR plays a crucial role in plant growth under mild drought, through the analyses with *Arabidopsis phr1 phl1* mutant deficient in PSR. (Fig. 6; Supplementary Fig. 16). Our results with soybean and *Arabidopsis* also show that mild drought inhibits Pi uptake (Supplementary Fig. 15), but that leaf water content is maintained to some extent (Supplementary Fig. 13d), and that Pi concentration in the leaves is reduced (Supplementary Fig. 13e, 14d), thus inducing PSR in plants (Fig. 4, 5; Supplementary Fig. 14i). These results are consistent with the previous observations that reduced soil moisture limits Pi diffusion in the soil^{43,47}. Under severe drought conditions, we show that PSR induction is suppressed by a relative increase in Pi concentration (Supplementary Fig. 13e) due to a decrease in leaf water content (Supplementary Fig. 13d). By contrast, ABA cannot respond in situations where some water is maintained in the leaf but responds when the leaf is losing water (Supplementary Fig. 13d, f) and is increasing the osmotic potential⁴⁸; as a result, the PSR occurs prior to the ABA response (Fig. 4e)."

COMMENT 22:

9. Line 339: this is why your analysis should be based on water potential, not soil water content.

RESPONSE 22:

Thank you for your valuable comment. We are aware that there are arguments that water potential-based descriptions are useful (e.g., Juenger and Verslues, Plant Cell 2023). On the other hand, even though such claims have been made for many years, the idea has not always been widely accepted, probably because of the high threshold for measuring water potential and technical difficulties such as large errors due to the conversion of qualitatively different values. In the future, we hope that these problems will be resolved. At this time, to facilitate and satisfy the understanding of a wider range of readers, we have included water potential in addition to soil moisture content in this paper (Lines 168, 171, 206, 263–265, 293–294, 754–757; Figs. 1, 4–6; Supplementary Figs. 7–9, 13, 14), as we described in RESPONSE 11, RESPONSE 18, and RESPONSE 19.

COMMENT 23:

10. Line 350-354: the authors over-hype the usefulness of ridges, which are not likely to work in areas with different rainfall frequencies or soil type.

RESPONSE 23:

Thank you for pointing this out. In response to the reviewer's suggestion, we have changed the sentence by adding the following statement (Lines 421–422):

“, albeit vary in degree depending on rainfall and soil type,...”

COMMENT 24:

11. There are other transcriptome data sets of plants subjected to moderate low water potential stress (for example, the Des Marias et al study that is cited, as well as others). The authors should compare their transcriptome data to these other data sets to see how much overlap there is. Has this enrichment of phosphate starvation response genes occurred in other studies and been overlooked, or interpreted in a different manner compared to here? If it was not observed in other studies, what are possible reasons for this?

RESPONSE 24:

Thank you for your interesting comment. We had mentioned this in the previous version of our manuscript, but have further modified the text (Lines 366–373) in response to your comment as follows:

"With the ultimate aim of minimizing crop loss due to insufficiently optimized water supply, this study focused on mild drought stress that is not severe enough to cause the leaves to wilt, in the field, where available P is not abundant and P acquisition is dependent on the biotic and abiotic environment⁴³. Previous papers reporting transcriptome analyses of soybean and *Arabidopsis* plants exposed to mild drought stress were unable to detect PSR behavior because, based on leaf wilting and water potential values, these analyses used more severe drought stress conditions and/or more nutrient-rich conditions in commonly used commercial culture soil than our experiments^{7-11, 44-46}."

Reviewer #1 (Remarks to the Author):

The revised version of the manuscript titled 'Phosphate starvation response precedes abscisic acid response under progressive mild drought in plants' has shown significant improvement compared to the first submission. Although two of my comments on inositol phosphate measurements were not addressed, I believe that the main message conveyed in the manuscript holds significant importance.

Reviewer #2 (Remarks to the Author):

The authors have done much extra work to respond to previous concerns. I have no further concerns to raise and commend the authors on their very thorough work.

Response to the comments made by the Editor and Reviewers

NCOMMS-22-35505B

Phosphate starvation response precedes abscisic acid response under progressive mild drought in plants

** If you wish to forward this email to your co-authors, please delete the link below to your author home page **

Dear Dr Fujita,

Your manuscript entitled "Phosphate starvation response precedes abscisic acid response under progressive mild drought in plants" has now been seen again by our referees, whose comments appear below. In light of their advice I am delighted to say that we are happy, in principle, to publish a suitably revised version in Nature Communications under the open access CC BY license (Creative Commons Attribution v4.0 International License).

We therefore invite you to revise your paper one last time to address the remaining concerns of our reviewers. At the same time we ask that you edit your manuscript to comply with our format requirements and to maximise the accessibility and therefore the impact of your work.

RESPONSE to the Editor:

Thank you so much for the opportunity to improve our paper for publication in Nature Communications. We have prepared the manuscript files according to all the instructions you have given us and are now resubmitting it.

The changes, other than those requested on the editorial, are as follows:

Lines 6, 25, and 26: We have added the author's present address.

Lines 671, 673 and 676: We have added the version information of the software.

Lines 769: We have added information on rhizobia that was missing from the previous version.

In the uploaded Supplementary Information, the changes, other than those requested on the editorial, are as follows:

We have added a cover of the file and fine-tuned all the figures and legends in terms of presentation.

Supplementary Figs. 2 and 3: We have fine-tuned the angle of the compass to the actual situation.

Supplementary Fig.7 and 8: We have excluded unnecessary " $P < 0.05$ " in the legend.

Supplementary Fig.12: We have excluded unnecessary " $P < 0.05$ " and added " $P < 0.001$ " in the legend.

REVIEWERS' COMMENTS

Reviewer #1 (Remarks to the Author):

The revised version of the manuscript titled 'Phosphate starvation response precedes abscisic acid response under progressive mild drought in plants' has shown significant improvement compared to the first submission. Although two of my comments on inositol phosphate measurements were not addressed, I believe that the main message conveyed in the manuscript holds significant importance.

RESPONSE to the Reviewer #1

Thank you so much for your valuable comments and suggestions to improve our manuscript.

Reviewer #2 (Remarks to the Author):

The authors have done much extra work to respond to previous concerns. I have no further concerns to raise and commend the authors on their very thorough work.

RESPONSE to the Reviewer #2

We sincerely appreciate your positive feedback on our additional research. Your valuable comments and suggestions have greatly helped us to improve the manuscript.